

# Disentangling the drivers of future Antarctic ice loss with a historically-calibrated ice-sheet model

Violaine Coulon[1], Ann Kristin Klose[2,3], Christoph Kittel[4], Tamsin Edwards[5], Fiona Turner[5], Ricarda Winkelmann[2,3], and Frank Pattyn[1]

[1]Université libre de Bruxelles (ULB), Laboratoire de Glaciologie, Brussels, Belgium
[2]Potsdam Institute for Climate Impact Research (PIK), Member of the Leibniz Association, P.O. Box 6012 03, D-14412 Potsdam Germany
[3]Department of Physics and Astronomy, University of Potsdam, Potsdam, Germany
[4]Institut des Géosciences de l'Environnement (IGE), Univ. Grenoble Alpes/CNRS/IRD/G-INP, Grenoble, France
[5]King's College London, Department of Geography, United Kingdom of Great Britain – England, Scotland, Wales

**Correspondence:** Violaine Coulon (violaine.coulon@ulb.be)

**Abstract.** We use an observationally-calibrated ice-sheet model to investigate the future trajectory of the Antarctic ice sheet related to uncertainties in the future balance between sub-shelf melting and ice discharge on the one hand, and the surface mass balance on the other. Our ensemble of simulations, forced by a panel of CMIP6 climate models, suggests that the ocean will be the primary driver of short-term Antarctic mass loss, initiating ice loss in West Antarctica already during this century. The atmosphere initially plays a mitigating role through increased snowfall, leading to an Antarctic contribution to global mean sea-level rise by 2100 of 6 (-8 to 15) cm under a low-emission scenario and 5.5 (-10 to 16) cm under a very high-emission scenario. However, under the very high-emission pathway, the influence of the atmosphere shifts beyond the end of the century, becoming an amplifying driver of mass loss as the ice sheet's surface mass balance decreases. We show that this transition occurs when Antarctic near-surface warming exceeds a critical threshold of +7.5°C, at which the increase in surface runoff outweighs the increase in snow accumulation, a signal that is amplified by the melt–elevation feedback. Therefore, under the very high-emission scenario, oceanic and atmospheric drivers are projected to result in a complete collapse of the West Antarctic ice sheet along with significant grounding-line retreat in the marine basins of the East Antarctic ice sheet, leading to a median global mean sea-level rise of 2.75 (6.95) m by 2300 (3000). Under a more sustainable socio-economic pathway, we find that the Antarctic ice sheet may still contribute to a median global mean sea-level rise of 0.62 (1.85) m by 2300 (3000). However, the rate of sea-level rise is significantly reduced as mass loss is likely to remain confined to the Amundsen Sea Embayment, where present-day climate conditions seem sufficient to commit to a continuous retreat of Thwaites Glacier.

## 1 Introduction

The Antarctic ice sheet (AIS) holds the largest amount of grounded ice on Earth, representing about 58-m sea-level equivalent (Fretwell et al., 2013; Morlighem et al., 2019). It is therefore the largest potential contributor to future sea-level rise. During the last few decades, the AIS has contributed to about 10% of the observed sea-level rise (Fox-Kemper et al., 2021). However, current observations indicate a growing rate of the Antarctic contribution to sea-level rise (Micheal et al., 2019). More





specifically, recent observations (e.g., Shepherd et al., 2018; Rignot et al., 2019) reveal that, since the early 2000s, the AIS has been losing mass at an accelerating rate. This mass loss primarily occurs along the ice sheet's periphery due to increased glacier flow. The observed acceleration of Antarctic outlet glaciers (essentially in West Antarctica and in some sectors of the

East Antarctic ice sheet; Rignot et al., 2019) is concentrated in areas close to warm and salty circumpolar deep water (CDW), which induce strong sub-shelf melt rates. The resulting thinning of the floating ice shelves reduces their ability to restrain the ice flowing from the grounded ice sheet towards the ocean (called buttressing effect), therefore destabilising the glaciers and raising sea level by increased ice discharge (Fürst et al., 2016; Gudmundsson et al., 2019; Reese et al., 2018b). Conversely, there is no clear continent-wide long-term trend in snowfall accumulation in the interior of the ice sheet (Medley and Thomas,

2019; Kim et al., 2020).

Despite the relatively good understanding of the drivers of current Antarctic mass changes, projections of the future evolution of the Antarctic ice sheet are associated with large uncertainties. Specifically, whether the ice sheet is going to gain or lose mass by the end of the century remains unclear, especially under very high-emission scenarios (Seroussi et al., 2020; Edwards et al., 2021). This uncertainty may notably be explained by the remaining unknowns in the long-term impacts of competing

processes expected to increase in a warming climate (i.e., ocean-induced sub-shelf melt, snow accumulation, and surface runoff) and their modulation of the future trajectory of the Antarctic ice sheet. With rising atmospheric temperatures, future accumulation is expected to increase as a result of enhanced snowfall associated with higher saturated vapour pressure (Frieler et al., 2015). While surface runoff is currently a relatively minor contributor to mass loss in Antarctica (Lenaerts et al., 2019), with stable surface melt rates since 1979 (Munneke et al., 2012), it is likely to significantly compensate for mass gain through

snowfall under very high-emission scenarios (Kittel et al., 2021). This increase in surface runoff may be expected due to the exponential relationship between the air temperature and the surface melt (Trusel et al., 2015; Kittel et al., 2021; van Wessem et al., 2023). Since the Antarctic surface mass balance (SMB, the balance of accumulation through precipitation and ablation through erosion, sublimation and runoff at the ice sheet surface) is, via the accumulation of snow, the only potential negative contributor to sea-level rise (Medley and Thomas, 2019), it has a strong potential in mitigating future AIS

mass loss projected to be driven by stronger basal melting of ice shelves over the course of this century (Seroussi et al., 2020). Furthermore, it is important to underline the likely indirect role of increased rainfall and surface melt in controlling the stability of Antarctic ice shelves, and thereby the overall stability of the ice sheet (Lenaerts et al., 2019; Bell et al., 2018). An increase in surface runoff over the floating ice shelves may lead to ice-shelf thinning (Kittel et al., 2021) or even trigger ice-shelf collapse due to hydrofracturing (Trusel et al., 2015; Pollard et al., 2015; Gilbert and Kittel, 2021; van Wessem et al., 2023), thereby

weakening the ice shelves and potentially reducing their buttressing effect on the upstream grounded ice-sheet flow. The future balance between these competing processes is still poorly known. One model (that includes hydrofracturing and marine ice cliff instability mechanisms; DeConto and Pollard, 2016; DeConto et al., 2021) predicts AIS mass loss in excess of one meter of global sea-level equivalent at the end of the 21st century (with multiple meters of potential additional sea-level rise in the centuries thereafter; DeConto and Pollard, 2016; DeConto et al., 2021) while other combinations of climate and ice sheet

models suggest AIS mass gain (Seroussi et al., 2020; Payne et al., 2021). The simulated evolution of the West Antarctic ice sheet, though it varies widely among models, seems to agree on mass loss in response to the predicted changes in oceanic





conditions. Projections are, however, less convergent on the future fate of the East Antarctic ice sheet. While most simulations from the ISMIP6 ensemble display a significant increase in SMB, outweighing the increased ice discharge under very high-emission scenario forcings (Seroussi et al., 2020), a recent estimate suggests a small positive contribution to sea-level rise from

the East Antarctic ice sheet (EAIS) at 2100, but with a wide range depending on the scenario (-4 to +22 cm for the 5th to 95th percentiles; Stokes et al., 2022).

Given the uncertainties in future Antarctic mass loss due to unknowns in the long-term impacts of basal melting and changes in SMB, we investigate here the future trajectory of the Antarctic ice sheet until the end of the millennium by considering uncertainties in SMB and ocean-induced melt processes. More specifically, using the ice-sheet model Kori-ULB (previously

called f.ETISh; Pattyn, 2017; Sun et al., 2020; Seroussi et al., 2019, 2020), we run an ensemble of simulations that accounts for key uncertainties in both ice–ocean and ice–atmosphere interactions, allowing to quantify how these uncertainties translate into uncertainties in the projected behaviour of the AIS. We perform a Bayesian calibration of our ensemble of ice-sheet model simulations by comparing the model results over the past decades with a series of estimates of regional net mass balance from the latest Ice Sheet Mass Balance Inter-comparison Exercise (IMBIE; Otosaka et al., 2023). This allows attributing

a higher predictive weight to model simulations that demonstrate skill at reproducing the observations (Ritz et al., 2015; Ruckert et al., 2017; Nias et al., 2019; Wernecke et al., 2020). The calibrated projections extend to the end of the millennium using atmospheric and oceanic projections inferred from a subset of models from the sixth phase of the Coupled Model Intercomparison Project (CMIP6) under low and very high emission scenarios.

## 2 Methods

### 2.1 Ice sheet model and climate forcing

Due to the computational constraints of coupled ice sheet–ocean–atmosphere models (e.g., Siahaan et al., 2022; Pelletier et al., 2022), especially in the case of ensemble modelling, we project changes in Antarctic future mass balance by forcing a standalone ice-sheet model with independent atmospheric and oceanic boundary conditions provided by climate models. Specifically, we conduct simulations of the response of the AIS to environmental and parametric perturbations with the Kori-

ULB model. All simulations are performed at a spatial resolution of 16 km. Given that outputs from regional climate models (RCMs) downscaling projections from global climate models (GCMs) are not yet available on timescales beyond the end of the century, we compute future Antarctic SMB based on GCM outputs directly (similar to, e.g., Nowicki et al., 2020; Seroussi et al., 2020). However, this involves several drawbacks related to their coarse resolution and their low sophistication in representing important physical processes of polar regions (Kittel et al., 2021). In addition, the atmospheric fields projected

by climate models are derived under the assumption of a static present-day ice-sheet geometry. Because of the so-called melt-elevation feedback (i.e., ice-sheet mass loss and hence surface elevation lowering leads to warmer near-surface air temperature, causing further melting and ice loss; Levermann and Winkelmann, 2016), this leads to biases in the projected SMB, especially for long-term projections. For these reasons, instead of directly relying on the surface runoff fields derived by the GCMs, surface melt and runoff are here determined within the ice-sheet model using a simplified melt-and-runoff model capturing



the basic physical processes of refreezing versus runoff in the snow column (see Appendix B). Surface melt is calculated as a function of monthly air temperatures and precipitations by use of a positive degree-day (PDD) scheme (similar to previous studies such as DeConto et al., 2021; Golledge et al., 2019; Garbe et al., 2020). Surface runoff is then estimated by a simple thermodynamic parameterisation of the refreezing process (Janssens and Huybrechts, 2000). The melt–elevation feedback is incorporated through the use of a lapse rate correction of the air temperatures with changes in ice-sheet surface elevation (as in,

e.g., Bulthuis et al., 2019; Garbe et al., 2020; DeConto et al., 2021). In turn, near-surface air temperatures alter the amount of rainfall as well as the surface meltwater production and refreezing (see Appendix B), so that the evolving ice-sheet topography dynamically alters the SMB (computed as precipitation minus evaporation minus runoff). For resolving sub-shelf processes and generating basal melt rates, we rely on physically-based parameterisations that approximate the local thermal forcing based on far-field ocean properties provided by the GCMs (Asay-Davis et al., 2017). These parameterisations vary from simple functions

of ocean temperature (Jourdain et al., 2020; Favier et al., 2019; Burgard et al., 2022) to relatively complex parameterisations developed more recently from box and plume models (Reese et al., 2018a; Lazeroms et al., 2018, 2019).

  Our ice-sheet simulations begin in the year 1950 CE to allow comparisons with observations over the satellite era. Ice-sheet initial conditions are provided by an inverse simulation nudging towards present-day ice-sheet geometry (following Pollard and DeConto, 2012b; Bernales et al., 2017). More information on the ice-sheet model setup and initialisation is provided in

Appendix A. Hindcasts of the behaviour of the AIS over the period 1950-2014 are produced using changes in oceanic and atmospheric boundary conditions derived from the CMIP5 climate model NorESM1-M (Bentsen et al., 2013), which has been evaluated as well performing over the historical period around Antarctica (Barthel et al., 2020). As of the year 2015, changes in atmospheric and oceanic properties derived from a subset of CMIP6 climate models (MRI-ESM2-0, IPSL-CM6A-LR, CESM2-WACCM and UKESM1-0-LL) are used as forcing until the year 2300. After 2300, the climate is kept constant in time, allowing

to investigate the long-term impacts of early-millennium warming (often called sea-level commitment; Golledge et al., 2015). Given the limited amount of CMIP6 models providing projections until the year 2300, GCMs were only selected by their availability. The forcing applied is derived from both the Shared Socioeconomic Pathways (SSP) 5-8.5 and 1-2.6 scenarios, to estimate the possible long-term (multi-centennial to millennial) ice-sheet response to a very wide range of climate forcing. When adressing the uncertainty associated with CMIP climate models and scenarios, the direct application of atmospheric

and oceanic properties as a boundary condition for ice-sheet models would require generating a new initial state for each GCM. Therefore, climate forcing is instead implemented via the use of anomalies. Atmospheric forcing is derived in the form of monthly-averaged air temperature and sublimation anomalies as well as precipitation ratios (to avoid 'negative' absolute precipitation; Goosse et al., 2010) calculated with respect to the 1995-2014 mean seasonal variations. On the other hand, oceanic forcing is derived in the form of yearly-averaged temperature and salinity anomalies compared with the 1995-2014

mean. Missing values for the oceanic forcing on the continental shelf (due to the coarse resolution of GCMs) and in currently ice-covered regions are filled following Kreuzer et al. (2021). These anomalies are then added to reference fields (referred to as the present-day atmosphere and ocean climatologies) that are used as a baseline in the ice-sheet model, similar to the approach of Seroussi et al. (2019, 2020). This choice assumes that the fluctuations in the oceanic and atmospheric anomalies across years are more significant than the differences between the current climatic conditions of these fields in the CMIP GCMs and



the present-day climatologies that are employed (Nowicki et al., 2020). The GCM-derived time-evolving and spatially-varying atmospheric and ocean forcings are applied to quasi-equilibrated initial ice-sheet states.

## 2.2 Ensemble design and calibration

To explore the uncertainty in ice–ocean and ice–atmosphere interactions, we design a perturbed parameter ensemble including nine key parameters that govern processes at the ice–ocean–atmosphere boundary. These parameters and their considered
uncertainty ranges are summarised in Table 1. We use a Latin hypercube sampling to create 100 distinct parameter vectors, thereby producing a 100-member ensemble. Note that according to Loeppky et al. (2009), the number of samples for a sufficient level of accuracy should be about 10 times the input dimension (here 9).

The designed ensemble includes two types of parameters: discrete inputs and continuous parameters. The first three parameters from our parameter space are discrete inputs that account for initial state and climate forcing uncertainties. Specifically,
we account for uncertainty in the representation of present-day Antarctic climate (i.e., the applied present-day climatologies $\text{CLIM}_{\text{atm}}$ and $\text{CLIM}_{\text{ocn}}$) by using SMB and air temperature conditions for the 1995-2014 period based on different polar-oriented RCMs, namely the Modèle Atmosphérique Régional (MARv3.11; Kittel et al., 2021) and the Regional Atmospheric Climate MOdel (RACMO2.3p2; van Wessem et al., 2018). Similarly, we use different ocean present-day temperature and salinity fields, based on either the observed properties of the Antarctic Shelf Bottom Water on the continental shelves from
Schmidtko et al. (2014) or the recent estimate of present-day, three-dimensional fields of temperature and salinity of the coastal ocean around Antarctica produced in Jourdain et al. (2020). To sample uncertainty in the imposed climate forcing, we use a subset of CMIP6 climate models (GCM): MRI-ESM2-0, IPSL-CM6A-LR, CESM2-WACCM and UKESM1-0-LL.

The remaining five parameters are continuous and capture uncertainties in ice–atmosphere and ice–ocean interactions. Uncertainty in the intensity of the melt-elevation feedback is taken into account by considering a range for the atmospheric lapse
rate $\gamma_{\text{atm}}$ (influencing the magnitude of air temperature changes with evolving ice-sheet elevation), which encompasses observational uncertainties (Martin and Peel, 1978; Magand et al., 2004). Additionally, we include uncertainties in the simulated surface melt and runoff by sampling uncertainty in the degree day factors for the melting of ice $K_{\text{ice}}$ and snow $K_{\text{snow}}$ (Hock, 2003, 2005; Braithwaite, 2008), and the thickness of the thermally-active layer influencing meltwater refreezing ($d_{\text{ice}}$; Cuffey and Paterson, 2010; Huybrechts and de Wolde, 1999; Reijmer et al., 2012, see Appendix B). To address uncertainties in
ice–ocean interactions (i.e., sub-shelf melting), we use five distinct basal melt parameterisations ($M_{\text{param}}$) of various levels of complexity (e.g., Jourdain et al., 2020; Favier et al., 2019; Burgard et al., 2022; Reese et al., 2018a; Lazeroms et al., 2019, see Appendix C). Furthermore, we consider uncertainties in the parameter (for each of the considered basal melt parameterisations) that modulates the effective ice–ocean heat flux $\Gamma_{\text{eff}}$, i.e., the sensitivity of ice-shelf melt to ocean thermal forcing.

Overall, we have designed our parameter space in order for parameter ranges to be as wide as physically plausible (Edwards
et al., 2019). After the 1950-2014 historical hindcast simulations, the 100-member ensemble is applied for both socio-economic pathways and under constant present-day (1995-2014 average) climate conditions. In addition, using the same 100-member ensemble, additional projections were produced, under SSP5-8.5 only, for the following specific experiments: (i) projections applying the atmospheric and oceanic forcings separately (by considering constant present-day conditions as of the year 2015),





(ii) projections neglecting the melt-elevation feedback (i.e., no lapse-rate correction of the near-surface air temperatures with changes in ice-sheet elevation), and (iii) projections including surface melt-driven hydrofracturing of the ice shelves (estimated following Pollard et al., 2015).

Starting from the prior probability distribution of the uncertain input parameters (assumed to follow a uniform distribution), we determine the posterior probability distribution using Bayes' theorem. As such, each ensemble member is assigned a likelihood score based on the differences (discrepancies) between the model outputs and a series of spatially-aggregated estimates of regional net mass balance from the IMBIE over the past decades (Otosaka et al., 2023, see Table 2). We use regionally and temporally aggregated constraints and assume that the model-observation discrepancies are therefore uncorrelated, to avoid the challenge of estimating covariances in the likelihood function (Nias et al., 2019, 2023). As suggested by Aschwanden et al. (2013), we favour rates of change metrics. To evaluate our ensemble independently of our projections, we select IMBIE estimates that correspond to our historical simulation, specifically those prior to 2015.

In Bayes' theorem, we can assume a multivariate Gaussian likelihood function between the observed and simulated mass balance estimates, whose mean values are given by the observations and variances are given by the sum of the observational and structural (approximating the "model uncertainties") errors. This likelihood function assumes the discrepancies between the observed and simulated values are (a) independent in both space and time and (b) normally distributed. For a given simulation $j$ in the ensemble characterised by a particular parameter vector $\theta$, the likelihood score $s_j$ is defined by

$$s_j = \exp\left(-\frac{1}{2}\sum_{i=1}^{N_{\mathrm{obs}}}\left(\frac{\mathrm{mod}_i^j - \mathrm{obs}_i}{\sigma_i}\right)^2\right), \tag{1}$$

where $N_{\mathrm{obs}}$ is the number of observational estimates, $\mathrm{obs}_i$ a given mass balance estimate, $\mathrm{mod}_i^j$ is the equivalent predicted output from the $j^{th}$ simulation of the ensemble, and $\sigma_i^2$ is the discrepancy variance given by $\sigma_i^2 = \left(\sigma_i^{\mathrm{obs}}\right)^2 + \left(\sigma_i^{\mathrm{mod}}\right)^2$, with $\sigma_i^{\mathrm{obs}}$ and $\sigma_i^{\mathrm{mod}}$ denoting the observational and model structural errors, respectively. Note that the multiplicative constant preceding the squared differences in the likelihood function can be dropped due to the normalisation performed later. Similar to, e.g., Nias et al. (2019, 2023), we estimate the structural error by multiplying the observational error, here by a factor of 10. This implies that our confidence in our ability to model reality is far lower than the ability to measure it (Ritz et al., 2015; Edwards et al., 2019; Nias et al., 2019). The choice in the magnitude of the structural error (and therefore discrepancy variance) was essentially made to avoid the scores being heavily weighted to a small number of ensemble members (Nias et al., 2023). In addition, given that the CMIP6 climate models included in the parameter space have no influence on the calibration (as the historical simulations were produced using the CMIP5 climate model NorESM1-M; see section 2.1), we also aimed for a roughly equivalent distribution of the weights across the four GCMs (see Figures S1 and S2). Other examples of assumptions to quantify the discrepancy variance may be found in, e.g., Ruckert et al. (2017); Ritz et al. (2015); Wernecke et al. (2020).

To create a weight $w$, the score for each ensemble member is then normalised,

$$w_j = \frac{s_j}{\sum s_j}. \tag{2}$$



**Table 1. Parameters governing ice–ocean and ice–atmosphere interactions along with their uncertainty ranges used in the uncertainty analysis.** Note that the parameters which control the effective ice–ocean heat flux relate to the considered sub-shelf melt parameterisation (more details are provided in appendix C). The parameter $\Gamma_{\mathrm{eff}}$ originally takes a value within the range of [0 - 1], as defined by the Latin hypercube sampling. It is then applied to the uncertainty range of the parameter associated with $M_{\mathrm{param}}$. For instance, for the $j^{th}$ simulation of the ensemble, if $M_{\mathrm{param}}^{j}$ is the local quadratic parameterisation $M_{\mathrm{quad}}$, a $\Gamma_{\mathrm{eff}}^{j}$ of 0.5 would correspond to a $\gamma_T$ of $5.5 \times 10^{-4}\,\mathrm{m\,s^{-1}}$.

| Parameter | Uncertainty range | | Units |
|---|---|---|---|
| Atmospheric present-day climatology ($\mathrm{CLIM_{atm}}$) | MARv3.11 (Kittel et al., 2021) | | - |
| | RACMO2.3p2 (van Wessem et al., 2018) | | - |
| Oceanic present-day climatology ($\mathrm{CLIM_{ocn}}$) | Jourdain et al. (2020) | | - |
| | Schmidtko et al. (2014) | | - |
| CMIP6 GCM climate forcing (GCM) | MRI-ESM2-0 | | - |
| | UKESM1-0-LL | | - |
| | CESM2-WACCM | | - |
| | IPSL-CM6A-LR | | - |
| Atmospheric lapse rate ($\gamma_{\mathrm{atm}}$) | $5-12$ | | $^{\circ}\mathrm{C\,km^{-1}}$ |
| Thickness of the thermally-active layer ($d_{\mathrm{ice}}$) | $0-15$ | | m |
| Degree day factor for the melting of ice ($K_{\mathrm{ice}}$) | $4-12$ | | $\mathrm{w.e.\,mm\,PDD^{-1}}$ |
| Degree day factor for the melting of snow ($K_{\mathrm{snow}}$) | $0-6$ | | $\mathrm{w.e.\,mm\,PDD^{-1}}$ |
| Sub-shelf melt parameterisation ($M_{\mathrm{param}}$) | PICO model ($M_{\mathrm{PICO}}$; Reese et al., 2018a) | | - |
| | Plume model ($M_{\mathrm{plume}}$; Lazeroms et al., 2019) | | - |
| | Local quadratic parameterisation ($M_{\mathrm{quad}}$; Favier et al., 2019; Burgard et al., 2022) | | - |
| | ISMIP6 nonlocal quadratic parameterisation ($M_{\mathrm{JD20}}$; Jourdain et al., 2020) | | - |
| | ISMIP6 nonlocal quadratic parameterisation including a dependency on the local slope ($M_{\mathrm{JD20s}}$; Jourdain et al., 2020) | | - |
| Effective ice–ocean heat flux ($\Gamma_{\mathrm{eff}}$) | $\gamma_T^{\star}$ in $M_{\mathrm{PICO}}$ | $0.1-10 \times 10^{-5}$ | $\mathrm{m\,s^{-1}}$ |
| | $C_d^{1/2}\Gamma_{TS}$ in $M_{\mathrm{plume}}$ | $1-10 \times 10^{-4}$ | - |
| | $\gamma_T$ in $M_{\mathrm{quad}}$ | $1-10 \times 10^{-4}$ | $\mathrm{m\,s^{-1}}$ |
| | $\gamma_0$ in $M_{\mathrm{JD20}}$ | $1-4 \times 10^{4}$ | $\mathrm{m\,yr^{-1}}$ |
| | $\gamma_0$ in $M_{\mathrm{JD20s}}$ | $1-4 \times 10^{6}$ | $\mathrm{m\,yr^{-1}}$ |



**Table 2. Observational constraints of Antarctic regional mass balance from the Ice Sheet Mass Balance Inter-comparison Exercise (IMBIE; Otosaka et al., 2023) used for the Bayesian calibration of the ensemble**.

| Region | Time Period | Value | Uncertainty | Unit |
|---|---|---|---|---|
| West Antarctic ice sheet | 1992-1996 | -37 | 19 | $Gt\,yr^{-1}$ |
| | 1997-2001 | -42 | 19 | $Gt\,yr^{-1}$ |
| | 2002-2006 | -64 | 20 | $Gt\,yr^{-1}$ |
| | 2007-2011 | -129 | 23 | $Gt\,yr^{-1}$ |
| East Antarctic ice sheet | 1992-1996 | -27 | 33 | $Gt\,yr^{-1}$ |
| | 1997-2001 | 21 | 32 | $Gt\,yr^{-1}$ |
| | 2002-2006 | 21 | 34 | $Gt\,yr^{-1}$ |
| | 2007-2011 | 19 | 36 | $Gt\,yr^{-1}$ |
| Antarctic Peninsula | 1992-1996 | -7 | 11 | $Gt\,yr^{-1}$ |
| | 1997-2001 | 2 | 11 | $Gt\,yr^{-1}$ |
| | 2002-2006 | -20 | 11 | $Gt\,yr^{-1}$ |
| | 2007-2011 | -21 | 12 | $Gt\,yr^{-1}$ |

Finally, the calibrated ensemble is then qualitatively assessed with a series of spatially-aggregated estimates of ice-sheet net mass balance, surface mass balance, sub-shelf melting, and iceberg calving fluxes from recent satellite- and modelling-based studies (see Figure 1 and Appendix D).

## 3 Results

### 3.1 Historical trends and influence of the calibration

The behaviour of the ensemble over the historical period is displayed in Figure 1. The calibrated ensemble reproduces the historical trends in good agreement with estimates of ice-sheet mass change and its drivers over the past decades (Appendix D), thereby providing the projections with more robustness. The spread of the posterior distribution is reduced compared with the prior distribution (Figure 1a–d), i.e., the observational constraints are effective in reducing uncertainty in the model hindcasts even though a large tolerance was assigned for model structural error. The obtained likelihood weights and the resulting posterior distributions of the parameter space are displayed in Figures S1–S2. Regions where mass change occurs during the historical period for the calibrated ensemble are shown in Figure 1e–f. Rapid grounded-ice mass losses are reproduced around the margins of the ice sheet, especially in the Amundsen Sea Sector (particularly pronounced in the Thwaites Glacier area) as well as in the Aurora Basin in East Antarctica (Figure 1e–f), presumably triggered by an increase in sub-shelf melting (Figure 1c). Indeed, substantial ice-shelf thinning is simulated in these areas, similar to observations over the past decades (Smith et al., 2020; Rignot et al., 2019).





**Figure 1. Evolution of the 100-member ensemble of simulations of the Antarctic ice sheet over the historical period (1950-2014).**
Evolution of the Antarctic ice-sheet net mass balance (considering volume above flotation only), i.e., the rate of mass change contributing to
sea-level rise (a), surface mass balance (b), the sub-shelf melt fluxes (c), and dynamic ice loss, i.e., the calving fluxes (d) over the historical
period and comparison with observations (red lines - shaded area represent the uncertainty of the observations, shown as $\pm\,1.64\sigma$). Dashed
lines and pale blue-shaded areas represent the ensemble prior distributions (medians and 5–95% probability intervals) while solid lines and
dark blue-shaded areas represent the posterior (Bayesian calibrated medians and 5–95% probability intervals) distributions. Observations
and their uncertainty are listed in Appendix D. The spatial pattern of historical mass change over the ensemble is illustrated by the Bayesian
calibrated mean thickness change (e) and rate of elevation change (f) over the period 1950-2014. Black lines show the ensemble mean
grounding line position and grey lines show the ensemble mean calving front (allowed to evolve during the hindcast). The 100-member
ensemble of simulations is produced using Latin hypercube sampling in the parameter space defined in Table 1, sampling key uncertainties
in ice–ocean and ice–atmosphere interactions.





**Figure 2. Calibrated probabilistic projections of the evolution of the Antarctic ice sheet (AIS) until the end of the millennium.** Evolution of the AIS contribution to global-mean sea-level rise (calculated as in Goelzer et al., 2020) projected by the calibrated ensemble under constant present-day conditions and the shared socio-economic pathways (SSP) 1-2.6 and 5-8.5 until 3000 (a), with a focus on the period 2015-2300 (b) and this century (c). Solid lines and shaded regions show the medians and 5–95% probability intervals (N=100 per SSP scenario), with 5-year running average applied. Dashed lines show the median rate of contribution to global mean sea-level rise. Figures d–f represent the probability of being ungrounded under constant present-day climate conditions (d), SSP1-2.6 (e) and SSP5-8.5 (f) at different points in time throughout the millennium. For each scenario, the marginal probability of being ungrounded at a given point is computed using the Bayesian calibrated mean of the ensemble (N=100). Grey regions correspond to locations where there is a 0% probability of being ungrounded. Present-day grounding lines are shown in black.



## 3.2 The pattern of future Antarctic mass loss

The calibrated ensemble (posterior distributions) shows, with the exception of the first half of this century, a positive contribution to global mean sea-level rise (GMSLR) throughout the millennium (Figure 2a–c). The median rate of contribution to GMSLR becomes positive by around 2040 (Figure 2c). Until the end of the century, the projected sea-level change shows
no clear dependence on the emission scenario, with median contributions to GMSLR of 6 (-8 to 15 – 5–95 percentiles) cm under SSP1-2.6 and 5.5 (-10 to 16) cm under SSP5-8.5 (Figure 2c). The Antarctic contribution to GMSLR maintains a rate comparable to present day until a notable increase in the second half of the century (Figure 2c). This acceleration in ice loss is likely caused by the projected retreat of Thwaites Glacier, which is consistent across both emission scenarios (Figure 2d–f). Throughout the century, mass loss primarily occurs in the Amundsen Sea Embayment under both socio-economic pathways
(Figure 2d–f).

The pattern of mass loss under both emission pathways starts to diverge after the end of the century. While mass loss remains essentially limited to the Amundsen Sea Embayment until the end of the millennium under SSP1-2.6 (with some less likely exceptions; Figure 2e), the very high-emission scenario (SSP5-8.5) exhibits an acceleration of mass loss in the Amundsen Sea sector as well as the onset of grounding-line retreat in Siple Coast (Ross area). Their combination results in a collapse of
the West Antarctic ice sheet (WAIS) under SSP5-8.5, which is expected to be completed between 2300 and 2500 (Figure 2f). Additionally, early-millennium warming under the very high-emission pathway likely commits the East Antarctic ice sheet to grounding-line retreat in its marine basins, with a higher probability of retreat in the Wilkes basin than in the Aurora basin by the end of the millennium (Figure 2f). Altogether, these differences in the pattern of long-term mass loss result in diverging trajectories of the projected contributions to global-mean sea level beyond the current century, with significantly more mass
loss projected under SSP5-8.5 than SSP1-2.6 over the following centuries (Figure 2a,b): the contribution to GMSLR amounts to 2.75 (0.46 to 4.52) m by the year 2300 and 6.95 (2.40 to 13.47) m by 3000 under SSP5-8.5, compared with 0.62 (-0.26 to 1.56) m and 1.85 (-0.73 to 2.90) m under SSP1-2.6 by 2300 and 3000, respectively (Figure 2a). Interestingly, a continued retreat of Thwaites Glacier throughout the millennium is also predicted by the calibrated ensemble under constant present-day climate conditions (Figure 2d), leading to a median sea-level contribution of 2 (-6 to 9) cm by the end of this century, 0.20
(-0.17 to 1.21) m by 2300, and 1.32 (-0.4 to 2.1) m by the end of the millennium.

Overall, the spread in the sea-level projections increases with time, especially under the very high-emission scenario. The differences between the prior and posterior distributions are shown in Figure S3. The spread of the posterior distribution is reduced compared with the prior distribution, similar to the historical period, particularly at shorter timescales.

## 3.3 Drivers of Antarctic mass change

As our calibrated ensemble has demonstrated a good agreement with satellite- and modelling-based estimates of each of the contributors to Antarctic mass changes (i.e., surface mass balance, sub-shelf melting, and iceberg calving; see Figure 1) over the past decades, we now use it to analyse the projected evolution of the drivers of future Antarctic ice loss over the coming centuries. Similar to the observations over the past decades (Rignot et al., 2019), the projected Antarctic ice loss until the end

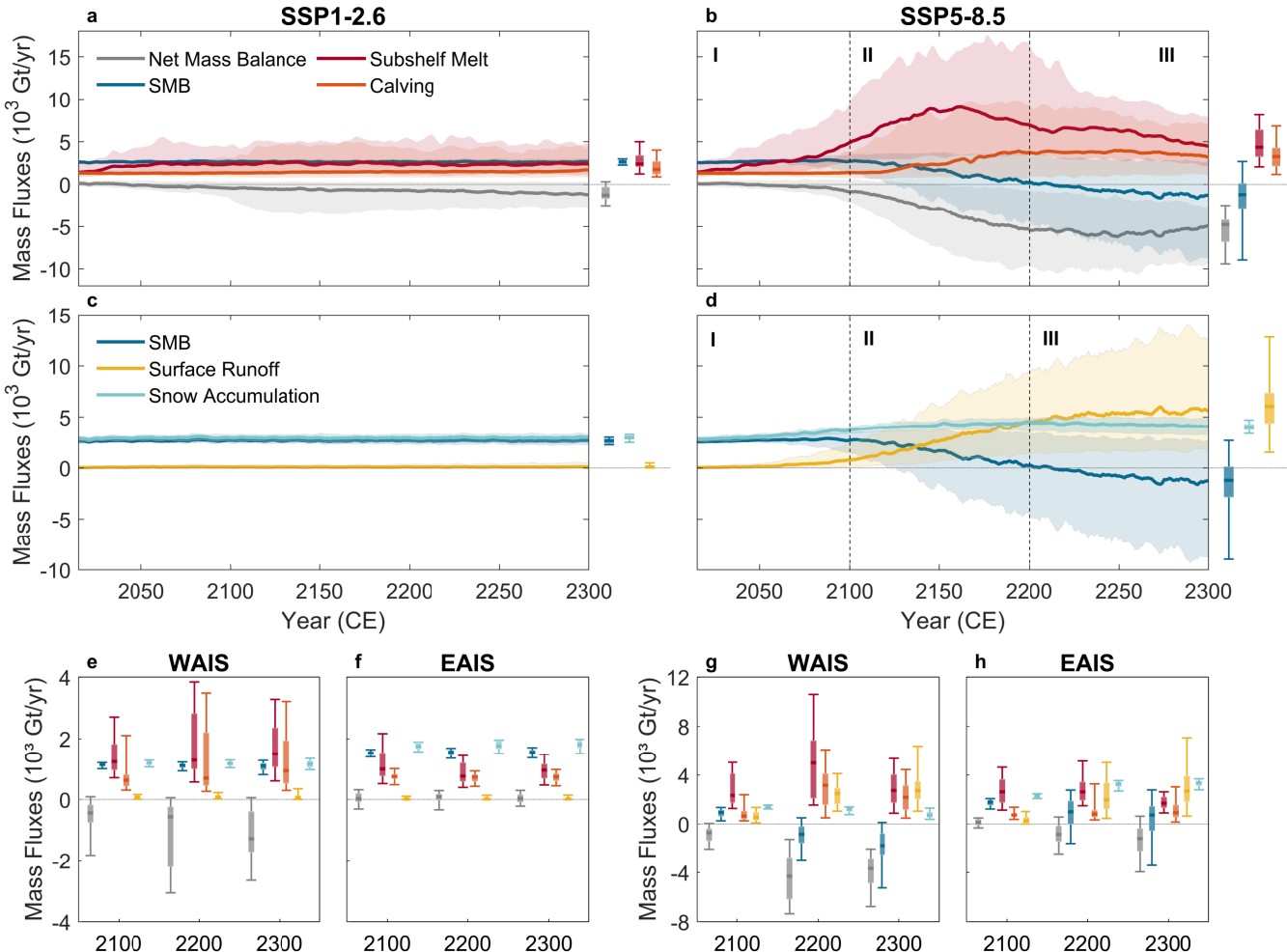

**Figure 3. Calibrated probabilistic projections of the Antarctic ice sheet mass balance components until the year 2300**. Evolution of the ensemble projected main ice sheet mass balance components (a–b) and surface mass balance components (c–d) for the 2015-2300 period under SSP1-2.6 (a,c) and SSP5-8.5 (b,d). Solid lines and shaded regions show the medians and 5-95% probability intervals (N=100 per SSP scenario), with 5-year running average applied. Boxes and whiskers show [5,25,50,75,95] percentiles at year 2300. Positive SMB fluxes represent mass gains while positive sub-shelf melt and calving fluxes represent mass losses. The boxes and whiskers in e–h show the evolution of these mass balance components in the West Antarctic ice sheet (WAIS; e,g) and in the East Antarctic ice sheet (EAIS; f,h) under SSP1-2.6 (e–f) and SSP5-8.5 (g–h) at years 2100, 2200 and 2300 (calculated as 5-year centered averages). Note that the ice-sheet net mass balance does not represent the sum of all mass balance components but instead considers changes in volume above flotation and may therefore be interpreted as the rate of mass change contributing to sea-level rise. Note also the changes in the y-axis ranges in the bottom figures between SSP1-2.6 (b–c) and SSP5-8.5 (e–f).




of the century is expected to be primarily driven by an increase in sub-shelf melt (Figure 3a–b), leading to ice-shelf thinning.
This thinning, by reducing the restraining potential (buttressing) of the floating ice shelves on the upstream grounded ice-sheet flow, triggers an increase in the amount of ice discharged into the ocean (Gudmundsson et al., 2019), which contributes to global mean sea-level rise. While sub-shelf melt fluxes stabilise in the second half of this century under SSP1-2.6 (although still contributing to a long-term collapse of the Thwaites Glacier area; Figure 2e), they are projected to further increase under the very high-emission pathway during and after this century. Overall, the similar trajectories of sea-level change projected
under both emission scenarios throughout this century (Figure 2c) suggest that the more pronounced ocean-driven ice loss triggered under SSP5-8.5 is partly mitigated by an increase in snow accumulation with warmer air temperatures (Figure 3c–d). Consistent with previous studies (e.g. Seroussi et al., 2020; Edwards et al., 2021), we find opposing sensitivities between the West and East Antarctic ice sheets over the 21st century: ocean-driven mass loss dominates the WAIS mass changes while East Antarctica is undergoing a (median) mass gain due to a dominating increase in SMB (Figure 3e–h and Figure S4).

While the increase in ice loss remains relatively stable under the low-emission pathway beyond the 21st century, the rate of contribution to GMSLR further increases (i.e., the net mass balance decreases) under SSP5-8.5, in conjunction with a decrease in SMB as well as an increase in calving fluxes (Figure 3b,d). The decrease in SMB occurs when the increase in surface runoff outweighs the increase in snow accumulation (Figure 3d). While median projected sub-shelf melt fluxes reach their maximum around 2150 (as ice shelves progressively collapse), surface runoff continues to rise, reaching median fluxes
of almost 6000 gigatons per year by 2300 (Figure 3d). Therefore, the highest rates of contribution to GMSLR are projected during the 23rd century, when surface runoff compensates for snow accumulation. This increase in surface runoff is found over both the West and East Antarctic ice sheets (Figure 3g–h). In East Antarctica, the combined effect of oceanic and atmospheric drivers transforms the ice sheet into a median positive contributor to GMSLR after the 21st century (Figure 3h).

Overall, the evolution of the main drivers of Antarctic mass loss under a very high-emission socio-economic pathway can
be characterized by distinct phases (Figure 3b,d): (I) until the end of the century, the acceleration in ice loss is primarily driven by the ocean, with a significant increase in sub-shelf melting that enhances the ice discharge by reducing the buttressing effect of the floating ice shelves; (II) after the end of this century, the net mass balance (contribution to sea-level changes) further decreases (increases) due to a decrease in SMB, thereby diminishing its mitigating effect, and (III) during the 23rd century, sub-shelf melt fluxes significantly decrease as ice shelves collapse, and Antarctic mass loss becomes dominated by the surface
mass balance. In contrast, mass loss under the low-emission pathway SSP1-2.6 may be characterised by the first phase only.

### 3.4 The relative importance of atmospheric and oceanic drivers of ice loss

To disentangle the relative influence of the atmospheric and oceanic drivers on projected mass loss under a very high-warming scenario, we produced, using the same parameter vectors (see section 2.2), additional calibrated projections for SSP5-8.5, isolating the different components of the climate forcing (Figures 4 and 5). More specifically, 100-member ensembles of
projections were produced for the following specific experiments:

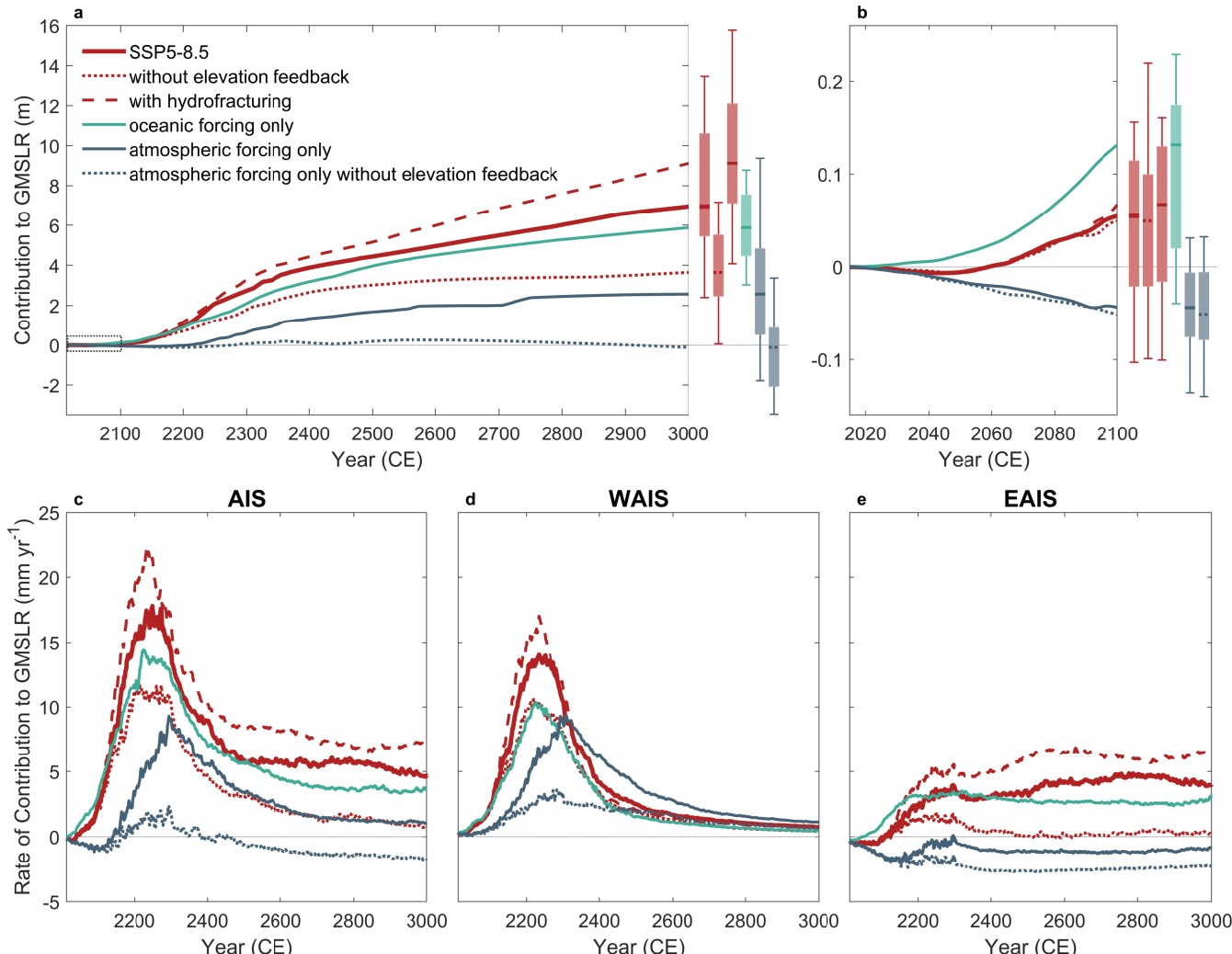

**Figure 4. Influence of oceanic and atmospheric drivers on the Antarctic contribution to future sea-level changes under the very high-emission pathway.** (a–b) Evolution of the calibrated median contribution to global mean sea-level rise (GMSLR) from Antarctica under the SSP5-8.5 scenario until 3000 (a) and focusing on the period 2015-2100 (b) for specific experiments (conducted with the 100-member ensemble) separating the respective influences of the oceanic and atmospheric forcings and of the melt-elevation feedback, and including the influence of ice-shelf hydrofracturing. The boxes and whiskers represent the 100-member ensemble posterior, indicating the [5,25,50,75,95] percentiles at 3000 (a) and 2100 (b) (N=100 per experiment). Figures c–e show the median rates of contribution to GMSLR for the whole Antarctic ice sheet (d), the West Antarctic ice sheet (e), and the East Antarctic ice sheet (d). A 5-year running average is applied.



**Figure 5. Contribution of atmospheric and oceanic forcings to projected Antarctic mass changes under the very high-emission pathway.** Mean ice thickness change at different points in time throughout the millennium under the shared socio-economic pathways (SSP) 5-8.5 using the 100-member ensemble for specific experiments considering (a) both oceanic and atmospheric drivers together, (b) oceanic forcing only, (c) atmospheric forcing only, and (d) atmospheric forcing only without the melt-elevation feedback. For each, the mean thickness change at a given point is computed using the Bayesian calibrated mean of the ensemble (N=100 per experiment). Black and grey lines show the ensemble mean grounding line and calving front positions, respectively.





- including the atmospheric forcing only while keeping the oceanic forcing constant (i.e., oceanic boundary conditions are kept constant to the present-day climatology as of the year 2015);

- including the oceanic forcing only while keeping the atmospheric forcing constant;

- including both the atmospheric and ocean forcings, but neglecting the melt-elevation feedback (i.e., no lapse-rate correction of the near-surface air temperatures with changes in ice-sheet elevation)

- including the atmospheric forcing only, while neglecting the melt-elevation feedback.

Overall, our sensitivity analyses confirm that projected Antarctic mass loss during the 21st century is primarily driven by the ocean. During this period, oceanic forcing alone produces the highest median Antarctic contribution to GMSLR (Figure 4b), with higher and earlier rates of contribution to GMSLR than when considering both oceanic and atmospheric drivers together (Figure 4c). On the other hand, atmospheric forcing alone initially leads to mass gain (negative median rates of contribution to GMSLR – Figure 4b–c), attributed to increased snow accumulation while surface runoff rates remain limited. This indicates that the atmospheric forcing mitigates ocean-driven Antarctic mass loss and sea-level rise during the 21st century under a very high-emission scenario. Our results also confirm the contrasting sensitivities of the West and East Antarctic ice sheets throughout the 21st century (Seroussi et al., 2020; Edwards et al., 2021). In the absence of the dominating increase in SMB, ocean forcing alone would transform the EAIS into a positive contributor to sea-level rise already in the second half of this century (Figure 4e). Conversely, in West Antarctica, the increase in snow accumulation would not be sufficient to prevent the ocean-driven retreat triggered during the historical period (Figures 4d and 5c).

Beyond the 21st century, as the increase in surface runoff surpasses the increase in snow accumulation, median GMSLR rates projected under atmospheric forcing alone start to increase and eventually become positive (around 2150; Figure 4c). This leads to a higher contribution to sea-level change when accounting for both oceanic and atmospheric drivers compared with oceanic forcing alone. Interestingly, we find that both the atmospheric and the oceanic forcings, when considered separately, are sufficient to trigger a complete collapse of the WAIS by the end of the millennium (Figure 5). While the retreat is delayed under atmospheric forcing alone, both forcings lead to rates of mass loss in West Antarctica reaching approximately 10 mm per year (Figure 4d). The high rates of mass loss projected under oceanic forcing alone may be attributed to the marine ice sheet instability mechanism (Weertman, 1974; Mercer, 1978). In contrast, the collapse of the WAIS and the associated high rates of contribution to GMSLR projected under atmospheric forcing alone are driven by the melt-elevation feedback (Figure 4 and Figure 5c–d). This suggests that while WAIS mass loss under SSP5-8.5 is initially triggered by the oceanic forcing (i.e., the increase in sub-shelf melting projected under SSP5-8.5 from the second half of this century; Figure 3b,g), its collapse is further accelerated by increased surface melt due to lowering surface elevations, which promotes the increase in surface runoff. In East Antarctica, retreat of the grounding line in the marine basins is initially triggered by the oceanic forcing, especially in the Wilkes basin (Figure 5a). While the signal and pattern of EAIS mass loss are primarily ocean-driven (Figures 5a and 4e), grounding-line retreat appears to be amplified by the influence of the melt-elevation feedback (compare Figures 5c–d). This pattern of committed mass loss in East Antarctica under SSP5-8.5 (Figures 2f and 5a) can be explained by the combination of





both climate forcings: the oceanic forcing triggers the deep inland penetration of the grounding line within the marine basins,
while the atmospheric forcing causes retreat close to the ice sheet margins, driven by the melt-elevation feedback (Figure 5a–
c). Without this feedback mechanism, surface runoff is significantly reduced and snow accumulation in the interior of the
ice sheet dominates the atmospheric signal, leading to significant mass gain in East Antarctica (Figure 5d), thus mitigating
mass loss. Therefore, despite the highly likely committed collapse of the WAIS driven by the melt-elevation feedback, the
calibrated ensemble under atmospheric forcing alone projects a significantly lower median sea-level contribution by the end of
the millennium (Figure 4a).

In summary, projected Antarctic mass loss under SSP5-8.5 is mainly driven by the combined effects of oceanic forcing and
the melt-elevation feedback. The oceanic forcing stands out as the primary trigger of mass loss in both West and East Antarctica.
The atmosphere initially mitigates this ocean-induced ice loss through enhanced snowfall. However, as we approach the end
of the century, the atmosphere transitions into an amplifying driver of mass loss, adding to the ocean-induced losses. This
amplifying effect is significantly strengthened by the melt-elevation feedback.

### 3.5    The influence of ice-shelf collapse

Given the observed correlation between the presence of melting at the surface of ice shelves and their collapse (Bassis and
Walker, 2012; Abram et al., 2013) and the high rates of surface runoff projected by our calibrated ensemble, we also investigated
the influence of the weakening of ice shelves by hydrofracturing (approximated following Pollard et al., 2015; DeConto and
Pollard, 2016) on the projected AIS mass loss under SSP5-8.5. It is important to note that our analysis focuses specifically on
hydrofracturing and does not include the marine ice cliff instability (MICI) mechanisms (Pollard et al., 2015; DeConto and
Pollard, 2016). Similar to Seroussi et al. (2020) and Pollard et al. (2015), we find that hydrofracturing produces higher median
rates of contribution to GMSLR, especially as of the end of the century, i.e., when surface runoff rates become significant
(Figure 4c–e). This leads to a median sea-level contribution increased by 60 cm at year 2300, and by 2.2 m at year 3000
compared with simulations that do not account for ice-shelf collapse through hydrofracturing (Figure 4a). The increase in the
rate of contribution to GMSLR arises from a significant acceleration of grounding-line retreat in West Antarctica as well as in
the marine basins of the EAIS. It is mainly explained by an acceleration of ice-shelf breakup, with a substantial increase of the
calving fluxes, which reach values of similar magnitudes as the sub-shelf melt in the second half of the 22nd century (Figure
S5).

### 3.6    Evolution of Antarctic surface mass balance in a warming climate

The mitigating influence of ice-sheet surface mass balance on sea-level rise is specifically related to the surface mass balance
over the grounded ice sheet. However, the stability of the ice sheet can be affected by interactions between the atmosphere and
the buttressing ice shelves, as highlighted in section 3.5. Figure 6a–b illustrate the changes in Antarctic surface mass balance
components over the grounded ice sheet and the ice shelves under the very high-emission socio-economic pathway SSP5-
8.5. While the median SMB over the floating ice shelves is expected to decrease in the second half of the 21st century, the
projected median SMB over grounded ice continues to increase until the end of the century (driven by an increase in snowfall).



**Figure 6. Evolution of Antarctic surface mass balance components over the grounded ice sheet and the ice shelves under a warming climate.** Figures a–b show calibrated probabilistic projections of Antarctic surface mass balance components (i.e., the total surface mass balance, snow accumulation, surface meltwater production, surface runoff, and rainfall) over the grounded ice sheet (a) and the ice shelves (b) under the shared socio-economic pathway (SSP) 5-8.5 until the year 2300 (with a zoom on the period 2015-2100 in b). Solid lines and shaded regions show the medians and 5–95% probability intervals (N=100 per SSP scenario), with 5-year running average applied. Boxes and whiskers show [5,25,50,75,95] percentiles at year 2300. Figures c–d show the sensitivity of future Antarctic surface mass balance over the grounded ice sheet (c) and the ice shelves (d) to Antarctic (90–60°S) near-surface warming. Yearly Bayesian calibrated median surface mass balance, snow accumulation and surface runoff ($\mathrm{Gt\,yr^{-1}}$) projected over the period 2015-2300 are compared with the mean annual near-surface temperature anomaly (in °C, with respect to the 1995-2014 average) projected by four GCMs from the sixth phase of the Coupled Model Intercomparison Project (CMIP6) over the Antarctic domain under the Shared Socioeconomic Pathways 1-2.6 and 5-8.5.





After 2100, the grounded-SMB starts to decrease (Figure 6a–b), along with its capacity to mitigate sea-level rise. In both cases, SMB decreases because of a substantial increase in surface runoff, associated with an increase in both surface melt and rainfall with rising air temperatures. By approximately 2140 (median; Figure 6a), the SMB over grounded ice becomes lower

than its present-day value. This means that the increase in SMB over the grounded ice sheet no longer offsets the increase in ice discharge driven by sub-shelf melting since 2015 (Figure 3). Despite the significant projected increase in surface runoff (reaching median fluxes of about 4000 Gt/yr by 2300; Figure 6a), SMB over the grounded ice sheet may remain positive. However, its mitigating potential approaches zero by around 2300 (median), when surface runoff fluxes are projected to fully offset snow accumulation (Figure 6a).

The aforementioned time evolution of the projected Antarctic (surface) mass balance strongly depends on the imposed climate forcing. To better understand the sensitivity of the surface mass balance and its components under a warming climate, we examine the relationship between the ensemble median SMB, snow accumulation and surface runoff and the regional (90–60°S) annual near-surface warming imposed by the GCMs (see Figure S6) over both the grounded ice sheet and the ice shelves (Figure 6c–d). We find a clear trend in the evolution of the Antarctic SMB under a warming climate. Consistent with

Kittel et al. (2021), Antarctic near-surface warming of more than +2.5°C (which may already be reached within this century, even under the low-emission socio-economic pathway; Figure S6) leads to a decrease in SMB over the ice shelves (Figure 6d), hence less efficiently mitigating the ocean-driven thinning of the buttressing ice shelves. Over the grounded ice sheet, the dominant factor in surface mass balance is the increase in snow accumulation until warming exceeds approximately +7.5°C, at which point the increase in surface runoff surpasses the increase in snow accumulation (consistent with previous findings

by Kittel et al., 2021). Beyond this threshold, which could be reached by the end of the century under SSP5-8.5 (Figure S6), the mitigating potential of surface mass balance starts to decrease. At the same level of near-surface warming (+7.5°C), surface runoff fluxes over the ice shelves are projected to fully offset the increase in snow accumulation, resulting in negative median surface mass balance over the ice shelves directly contributing to the weakening of their buttressing potential. Over the grounded ice sheet, SMB reaches its present-day value and hence no longer compensates for the increase in ice discharge

when the near-surface warming approaches +9°C. A significant threshold of negative surface mass balance (i.e., no longer offsetting any ice-sheet mass loss) is not projected until a strong Antarctic annual near-surface warming of +15°C (Figure 6c). For comparison, the Antarctic near-surface warming with respect to present-day (1995-2014 average) is compared with the pre-industrial (relative to the 1850-1900 period) global temperature change in Figure S7. The thresholds of +2.5°C, +7.5°C and +15°C of GCM-mean Antarctic near-surface warming relative to present-day correspond to a pre-industrial global warming of

approximately +3.0°C, +7.2°C and +12.2°C, respectively.

## 4  Discussion

We have produced observationally-calibrated projections of the evolution of the Antarctic ice sheet until the end of the millennium, accounting for key uncertainties in ice–climate interactions. Our projections of the future sea-level contribution from the Antarctic ice sheet by the end of this century relative to 2015 range from -8 to 16 cm for the 5–95% calibrated probability





interval. These estimates align with earlier Antarctic sea-level projections at 2100 relative to 2015 provided within the framework of the recent ice sheet model intercomparison ISMIP6 under climate forcings based on a subset of CMIP5 (ranging from -1 to 16 cm and from -8 to 30 cm under RCP2.6 and RCP8.5, respectively; Seroussi et al., 2020) and CMIP6 (ranging from -5 to 1 cm and from -9 to 11 cm under SSP1-2.6 and SSP5-8.5, respectively; Payne et al., 2021) climate models, as well as with the estimates obtained by emulating the ISMIP6 ensemble (-5 to 14 cm for the 5–95% probability interval under both SSP1-2.6

and SSP5-8.5; Edwards et al., 2021). Our calibrated projections also confirm previous findings that the emission scenario has a limited influence on Antarctic ice loss by the end of the century, as highlighted by Edwards et al. (2021) and Lowry et al. (2021). This lack of clear dependence on the emission scenario, along with the similar pattern of ice loss observed under both SSP1-2.6 and 5-8.5 over the 21st century, can be attributed to the fact that the ocean-driven retreat of Thwaites Glacier was already triggered during the historical period (as shown by Figure 2f).

By 2300, our projections indicate a range of sea-level rise from -0.26 to 1.56 m under SSP1-2.6 and from 0.46 to 4.52 m under SSP5-8.5, relative to 2015 (5–95% calibrated probability interval). These projections are again consistent with previous estimates (e.g., Lowry et al., 2021), including those provided in the latest assessment of the IPCC (–0.14 to 0.78 m SLE under RCP2.6/SSP1-2.6, and –0.27 to 3.14 m SLE under RCP8.5/SSP5-8.5; Fox-Kemper et al., 2021). The projections for the low-emission scenario are also in line with those by Turner et al. (2023), who combined four sets of projections from current

literature (Lowry et al., 2021; Levermann et al., 2020; Bulthuis et al., 2019; DeConto et al., 2021) and estimated a range of -0.1 to 1.5 m (5–95% probability interval) for SSP1-2.6 by 2300 compared to 2015. Notably, the projections by Bulthuis et al. (2019) – also derived using perturbed parameter ensembles and with a previous version of the same ice-sheet model – led to ranges of -0.14 to 0.47 m and 0.17 to 3.12 m GMSLR by 2300 under RCP2.6 and RCP8.5, respectively (outer 5% and 95% probability intervals across different sliding laws). We attribute the slightly lower sensitivity of their projections (also noted

by Turner et al., 2023) to a simplified climate forcing based on spatially-uniform air temperature changes and the absence of historical trends (Reese et al., 2020). In contrast, including hydrofracturing and the MICI mechanisms, DeConto et al. (2021) projected a significantly higher contribution of around 7–14 m to global mean sea-level rise by 2300 under a very high emission scenario.

    Our study also explored the committed long-term changes of the AIS in response to early-millennium warming, i.e., the mass

change that continues even after the climate forcing is held constant beyond the year 2300 (Figures 2, 4 and 5). Similar to the findings of Chambers et al. (2021), who evaluated the long-term impact of 21st-century warming, our projections reveal that West Antarctica experiences more severe ice loss than East Antarctica under the unabated warming path simulations. However, our projections indicate higher ranges of AIS contribution to sea-level rise. We attribute this difference to two factors: (i) the wider range of warming (i.e., post-2100) considered in our study and (ii) the absence of the melt-elevation feedback, which has

been shown to increase rates of mass loss (Figure 4), in the simulations conducted by Chambers et al. (2021). Furthermore, the long-term behaviour of our ensemble under constant present-day climate conditions aligns with Golledge et al. (2019), which showed that Thwaites Glacier retreats significantly with climatic boundary conditions held constant from 2020.

    Benefiting from our initialisation procedure, which generates an ice sheet that closely matches the observations (Figures S8–S9), we performed a qualitative evaluation of our calibrated ensemble by comparing it with spatially-aggregated estimates





of the main components of the ice-sheet mass balance (sub-shelf melt, calving, and surface mass balance fluxes; Appendix D) during the historical period (Figure 1). We then investigated the evolution of these drivers of future Antarctic mass changes while accounting for uncertainties in both ice–ocean and ice–atmosphere interactions. In agreement with recent studies (e.g., Golledge et al., 2019; Lowry et al., 2021), our projections indicate a divergence in the rate of ice loss between the different emission scenarios around 2060-2080 (Figure 2c). Similar to Lowry et al. (2021), this period marks the onset of accelerated ice-

sheet retreat under a very high-emission scenario, leading to a partial collapse of the WAIS by 2300. The substantial increase in the sub-shelf melt fluxes projected by our calibrated ensemble throughout the 21st century agrees with Golledge et al. (2019), with fluxes of about $5000 \, \mathrm{Gt \, yr^{-1}}$ reached by the end of the century. This period is also characterised by an increased SMB due to enhanced snow accumulation, again in agreement with other studies (e.g., Golledge et al., 2019; Siahaan et al., 2022). Similar to Kittel et al. (2021), we find that the SMB over grounded ice is projected to increase by the end of the century under

SSP5-8.5 as a response to stronger snowfall, only partly offset by enhanced meltwater runoff. On the other hand, over the ice shelves, strong runoff fluxes associated with higher temperatures are projected to decrease the SMB (Figure 6a–b). These results are consistent with the first two-way coupling of atmosphere and ocean models to a dynamic model of the Antarctic ice sheet until the end of the century by Siahaan et al. (2022), which therefore captures the melt-elevation feedback.

Over the coming centuries, as atmospheric warming and hence surface runoff keeps increasing, our results suggest a transi-

tion in the drivers of Antarctic mass loss, switching from ocean-driven to atmospheric-driven mass changes. Ultimately, under a very high-emission scenario, surface runoff is projected to dominate the Antarctic mass balance by the end of the 23rd century. Our results also indicate that the anticipated increase in surface runoff under a very high-warming scenario could accelerate Antarctic mass loss due to ice-shelf collapse triggered by hydrofracturing. However, it is important to note the absence of a water-routing scheme in our projections. Instead, similar to other studies (e.g., Seroussi et al., 2020; Pollard et al., 2015;

DeConto et al., 2021), we assume that surface meltwater that does not refreeze in winter is stored locally in crevasses, thereby potentially leading to ice-shelf collapse by hydrofracturing (Pollard et al., 2015). In reality, meltwater may also be transported laterally and exported to the ocean (Bell et al., 2017, 2018), reducing the likelihood of its contribution to hydrofracturing (Lai et al., 2020). Therefore, our projections may overestimate the risk of surface melt-induced destabilisation. Nonetheless, even when considering hydrofracturing processes, our projections remain significantly lower than projections of ice loss incorporat-

ing MICI mechanisms (Pollard et al., 2015; DeConto and Pollard, 2016; DeConto et al., 2021).

It is important to acknowledge that our approach does not account for potential feedback between ice and climate that may influence the climate forcing and subsequently mass loss, such as the influence of ocean meltwater discharge on atmospheric cooling (Golledge et al., 2019; DeConto et al., 2021; Purich and England, 2023). Considering such cooling feedback would counterbalance some of the atmospheric warming predicted by climate models and consequently delay the onset of high surface

runoff and its impact on AIS mass loss. However, recent studies suggest that the input of freshwater from ice shelves in the ocean is also likely to modify ocean conditions and may create a positive feedback loop by stratifying the water column and trapping warm water below the sea surface. This would result in ocean warming and increased melt rates near the grounding line around most Antarctic margins (Bronselaer et al., 2018; Golledge et al., 2019; Sadai et al., 2020; Li et al., 2023; Purich





and England, 2023). This mechanism, which is not included in most climate models projections, could therefore increase the
projected mass loss from the Antarctic ice sheet (Golledge et al., 2019).

It is also worth noting that the SSP5-8.5 scenario (used here along with the low-emission SSP1-2.6 scenario) is considered
very unlikely to be followed on these multi-century timescales (Hausfather and Peters, 2020; Schwalm et al., 2020). Therefore,
our projections under this very high-emission scenario should be interpreted as a maximum imagined outcome. In order to
represent a more plausible outcome, it would be necessary to generate Antarctic projections based on alternative scenarios,
such as SSP1-1.9 or intermediate pathways. Unfortunately, GCM projections for scenarios other than SSP1-2.6 and SSP5-
8.5 are not yet available beyond 2100. In order to reduce the dependence on the applied climate forcing, we examined the
relationship between median SMB estimates and regional atmospheric warming imposed by the GCMs (Figure 6c–d). This
analysis allowed us to identify and confirm potential thresholds in future Antarctic SMB. Notably, our results support the
findings from Kittel et al. (2021), suggesting the existence of a threshold at +7.5°C in near-surface warming over Antarctica
(which may already be reached by the end of this century under SSP5-8.5; Figure S6). This threshold results in reduced
SMB for the grounded ice sheet due to a significant increase in surface runoff. Establishing similar relationships for other ice
sheet mass balance components is, however, more challenging given that they are less straightforwardly linked to atmospheric
warming than SMB.

Given the influence of surface melt and runoff on projected mass loss, we assess the accuracy of our PDD-based melt-and-
runoff scheme by comparing our projections of surface runoff with outputs from both regional (MAR; Kittel et al., 2021)
and general (CESM2-WACCM; Dunmire et al., 2022) polar-oriented climate models under an SSP5-8.5 scenario (Figure 7).
Overall, our PDD-based projections of surface runoff demonstrate good agreement with climate models in terms of both pattern
(Figure 7a–f) and magnitudes (Figure 7g–h). Differences in runoff magnitudes can be attributed to changes in ice-sheet area
in our projections, as illustrated by the correction applied to the climate models' outputs (see Figure 7g–h). Notably, several
simulations in our ensemble display a partial collapse of the Larsen and George VI ice shelves by 2100 (Figure 7a), which
contributes to the lower median total runoff rates reproduced by the calibrated ensemble as compared with MAR in the last
decades of the 21st century. Although our projections account for the melt-elevation feedback (explaining higher runoff rates
in the Antarctic Peninsula and Thwaites Glacier regions at 2100 and in the WAIS and Wilkes basin at 2300), our PDD-based
model tends to underestimate aggregated surface runoff rates compared with climate models projections under high warming.
This discrepancy may be attributed to the omission of the melt-albedo feedback (Zeitz et al., 2021) in the PDD approach and
the coarse spatial resolution of the GCMs used for the climate forcing, which could contribute to the underestimation of the
runoff rates over the ice shelves. For a comprehensive comparison of projections of the other SMB components over both the
grounded ice sheet and the ice shelves with MAR outputs, please refer to Figure S10. Again, the main differences are primarily
influenced by changes in ice-sheet elevation (especially over the grounded ice sheet) and area (especially over the ice shelves).
Overall, we have produced credible projections of future Antarctic mass balance by (i) considering existing uncertainties and
(ii) conditioning ensemble projections on relevant observations (Aschwanden et al., 2021). In particular, we have performed an
exploration of uncertainties in model parameters with (due to the high computational demand of parameter exploration, espe-
cially at continental scale) a specific focus on ice–ocean–atmosphere interactions. However, the influence of other processes,



**Figure 7. Comparison of the surface runoff rates projected by the Bayesian calibrated ensemble under a very high-emission pathway to outputs from climate models.** Ensemble calibrated runoff fluxes in the year 2100 (a) and 2300 (e) compared with 2091-2100 average surface runoff projected by MAR forced by CESM2 (b), MAR forced by CNRM-CM6 (c) and CESM2-WACCM (d) and 2290-2299 average surface runoff projected by CESM2-WACCM (f) under the Shared Socio-economic Pathway (SSP) 5-8.5. In figures (a) and (e), black and grey lines show the ensemble mean grounding line and calving front positions, respectively. In figures (b–d) and (f), black lines show present-day grounding lines. Figures g–h show the evolution of the aggregated surface runoff over the period 2015-2100 (g) compared with projections from MAR forced by CESM2 (light grey line) and CNRM-CM6 (dark grey line), and over the period 2015-2300 (h) compared with projections from CESM2-WACCM. Orange solid lines and shaded areas show the ensemble calibrated median, 25-75% and 5–95% probability intervals (N=100 per SSP scenario), with 5-year running average applied. Crosses show aggregated fluxed reproduced by MAR (g) and CESM2-WACCM (h) over the ensemble calibrated mean mask (i.e., the total ice-sheet area) at different points in time.





which are known to have a significant influence on ice-sheet behaviour and are likewise characterised by significant uncertain-
ties, have not been explored here. Such processes include, amongst others, the sliding law used to determine basal shear stress
(Ritz et al., 2015), the influence of basal hydrology (Kazmierczak et al., 2022), or the influence of the spatial variability in
Antarctic viscoelastic properties, with its potential stabilising effect in the WAIS (Coulon et al., 2021; Whitehouse et al., 2019;
DeConto et al., 2021).

In addition, uncertainty quantification in our projections is intrinsically limited by the fact that they rely on a single ice-sheet
model, characterised by its uncertainties in boundary conditions, incomplete model physics, and choices in numerical methods
(Knutti and Sedláček, 2012; Williamson et al., 2014; Aschwanden et al., 2021), thereby neglecting structural uncertainty. Even
though our ensemble includes two different initial states (one for each present-day atmospheric climatology), they were derived
from the same initialisation procedure.

Forcing uncertainty has been sampled by applying multiple climate forcings for each emission scenario to drive the ice-sheet
model. Although the number of CMIP6 models providing projections until 2300 was limited, the selected GCMs encompass a
wide range of projected warming (Figure S6). We focused on state-of-the-art climate model projections from the sixth phase
of the Coupled Model Intercomparison Project (CMIP6), known to extend to warmer future climates compared with previous
model generations (Meehl et al., 2020). However, it is important to acknowledge that it is not clear whether CMIP6 models
have improved relative to the previous generation (Roussel et al., 2020; Jourdain et al., 2020; Fox-Kemper et al., 2021), and
that the higher climate sensitivity of some CMIP6 ESMs may not be supported by palaeo-climate records (Zhu et al., 2020).

Furthermore, it should be noted that the observational estimates used for calibrating and evaluating our projections only
consider spatially-integrated quantities, potentially obscuring unrealistic spatial trends (although the constraints used for the
Bayesian calibration were aggregated over three main Antarctic regions). While there is room for improvement, it is worth
highlighting that studies calibrating Antarctic sea-level projections have remained so far limited and have mainly focused on
simpler calibration techniques such as 'history matching' (e.g., DeConto et al., 2021; Lowry et al., 2021; Edwards et al., 2019;
Golledge et al., 2019). Only few have performed calibration in a Bayesian framework (Nias et al., 2019; Wernecke et al., 2020),
especially for continental scale projections (Ritz et al., 2015; Gilford et al., 2020; Ruckert et al., 2017).

Although calibration provides more robustness to projections, it is important to recognize that exploring additional sources of
uncertainties that have not been assessed here will likely alter and broaden the distributions of the projections presented in this
study (Edwards et al., 2021). Similarly, alternative choices for the magnitude of the structural error (and therefore discrepancy
variance; see section 2.2) would also influence our calibrated projections. As an illustration, estimating the structural error by
multiplying the observational error using a factor of 8 or 12 instead of 10 would change the main projection 5–95% intervals
by around ± 5% (Table S1). The sensitivity of the posterior distribution of the Antarctic sea-level contribution at various
discrepancy variances is shown in Figure S11.





## 5 Conclusions


We produced observationally-calibrated projections of the evolution of the Antarctic ice sheet until the end of the millennium. These projections are based on an ensemble of simulations that accounts for nine key uncertainties in ice–ocean–atmosphere interactions. Our results suggest that the ocean will be the primary driver of short-term mass loss, driving ice loss in the West Antarctic ice sheet already in the 21st century. Under a very high-emission scenario, we find that both the ocean and the

atmosphere, when considered independently, have the potential to lead to a complete collapse of West Antarctica by the end of the millennium. However, the combined effects of ice–ocean and ice–atmosphere interactions will likely cause WAIS collapse to occur earlier: initially triggered by the oceanic forcing, mass loss is then further accelerated by the melt-elevation feedback as near-surface warming increases. While the EAIS will, at first, experience mass gain due to increased snow accumulation, ocean-driven mass loss will take over as of the beginning of the next century under a very high-emission pathway, as the

mitigating role of SMB decreases. The decrease in SMB is associated with a strong increase in surface runoff with warming air temperatures, a signal significantly amplified by the melt-elevation feedback. More generally, this decrease in SMB seems to start when Antarctic near-surface warming exceeds +7.5°C, the threshold at which the increase in surface runoff outweighs the increase in snow accumulation. Reaching such a threshold may therefore be delayed under more sustainable socio-economic pathways. If this threshold is not crossed, i.e., under the socio-economic pathway SSP1-2.6, Antarctic mass loss would likely

remain limited to the Amundsen Sea Embayment region, where present-day climate conditions seem sufficient to commit to a continuous retreat of Thwaites Glacier. Overall, future Antarctic mass changes will likely be characterised by a transition in the primary driver of mass loss, shifting from an ocean-driven to an atmospheric-driven contribution of the AIS to GMSLR. The timing of this transition will be dictated by the trajectories of future atmospheric warming.

## Appendix A: Ice sheet model setup and initialisation

The Kori-ULB model is a vertically-integrated, thermomechanical, hybrid ice-sheet/ice-shelf model that incorporates essential characteristics of ice-sheet thermomechanics and ice-stream flow, such as the melt-elevation feedback, bedrock deformation, sub-shelf melting, and calving. The ice flow is represented as a combination of the shallow-ice (SIA) and shallow-shelf (SSA) approximations for grounded ice while only the shallow-shelf approximation is applied for floating ice shelves (Bueler and Brown, 2009; Winkelmann et al., 2011). In order to account for grounding-line migration, a flux condition (related to the ice

thickness at the grounding line; Schoof, 2007) is imposed at the grounding line following the implementation by Pollard and DeConto (2012a, 2020). This implementation has been shown to reproduce the migration of the grounding line and its steady-state behaviour (Schoof, 2007) at coarse resolution (Pattyn et al., 2013; Pollard and DeConto, 2020). Numerical simulations of the AIS using a flux condition have also been able to simulate marine ice-sheet behaviour in large-scale ice-sheet simulations (Pollard and DeConto, 2012a; DeConto and Pollard, 2016; Pattyn, 2017; Sun et al., 2020). While the use of such a flux

condition has been challenged, especially with respect to ice shelf buttressing and regimes of low driving and basal stresses (Haseloff and Sergienko, 2018; Pegler, 2018; Reese et al., 2018a; Sergienko and Wingham, 2019), Pollard and DeConto (2020) demonstrate that the algorithm gives similar results under buttressed conditions compared to high-resolution models.



Basal sliding is introduced as a Weertman sliding law, i.e., $v_b = -A_b |\tau_b|^{m-1}\tau_b$, where $\tau_b$ is the basal shear stress, $v_b$ the basal velocity, $A_b$ the basal sliding coefficient – whose values are inferred following the nudging method of Pollard and DeConto

(2012b) – and $m = 3$ a sliding exponent. Basal melting underneath the floating ice shelves may be determined by different sub-shelf melt parameterisation schemes, such as the PICO model (Reese et al., 2018a), the Plume model (Lazeroms et al., 2019), and simple parameterisations (Jourdain et al., 2020; Favier et al., 2019; Burgard et al., 2022). We employed data by either Schmidtko et al. (2014) or Jourdain et al. (2020) for present-day ocean temperature and salinity on the continental shelf. Calving at the ice front depends on the combined penetration depths of surface and basal crevasses, relative to total

ice thickness. The depths of the surface and basal crevasses are parameterised as functions of the divergence of ice velocity, the accumulated strain, the ice thickness, and (if desired) surface liquid water availability, similar to Pollard et al. (2015) and DeConto and Pollard (2016). Prescribed input data include the present-day ice-sheet geometry and bedrock topography from the Bedmachine dataset (Morlighem et al., 2019) and the geothermal heat flux by Shapiro and Ritzwoller (2004). Present-day mean near-surface air temperature and precipitation are obtained either from van Wessem et al. (2018), based on the regional

atmospheric climate model RACMO2.3p2, or from Kittel et al. (2021), based on the regional climate model MARv3.11. Near-surface temperatures are corrected for elevation changes using a vertical lapse rate (Pollard and DeConto, 2012a). Changes in bedrock elevation due to changes in ice load are modelled by the commonly used Elastic Lithosphere–Relaxed Asthenosphere (ELRA) model where the solid-Earth system is approximated by a thin elastic lithosphere plate lying upon a relaxing viscous asthenosphere (Brotchie and Silvester, 1969; Le Meur and Huybrechts, 1996). The viscoelastic properties of the Antarctic

solid Earth are considered as spatially-uniform and approximated using an asthenosphere relaxation time $\tau$ of 3000 years and a flexural rigidity of the lithosphere $D$ of $10^{25}\,\mathrm{N\,m}$.

Ice-sheet initial conditions and basal sliding coefficients are provided by an inverse simulation following Pollard and De-Conto (2012b), using surface mass balance forcing for the year 1950 (anomalies for the period 1945-1955 respective to the period 1995-2014 derived from CMIP5 NorESM1-M are added to a present-day climatology for the 1995-2014 period provided

by an RCM). In the inverse procedure, basal sliding coefficients under grounded ice, and sub-shelf melt rates under floating ice (Bernales et al., 2017) are adjusted iteratively to reduce the misfit with observed ice thickness (the calving front is kept to its observed present-day position). The obtained sub-shelf melt rates may therefore be regarded as the balance melt rates and are independent of the ocean boundary conditions (forcing). For consistency, different initial states are only produced for each atmospheric present-day climatology. Therefore, initial ice-sheet conditions (ice thickness, bed elevation, velocity, basal sliding

coefficients and internal ice and bed temperatures) are identical in all simulations that use the same present-day atmospheric climatology (either derived from RACMO2.3p2 or MARv3.11) and are in steady-state with the initial atmospheric boundary conditions. To limit an initial shock caused by the transition from the balance sub-shelf melt rates derived during the transient nudging spin-up to the imposed sub-shelf melt parameterisation scheme, a short 10-yr relaxation is run after the model initialisation, before the historical simulation, using constant atmospheric and oceanic forcings for the year 1950 (i.e., applying the

relevant atmospheric and oceanic present-day climatologies adjusted with 1945-1955 anomalies derived from NorESM1-M). Our initial states are therefore considered as quasi-equilibrated states. The two initialised ice sheet configurations resulting from the nudging spin-ups are within the range of the ISMIP6 models (Seroussi et al., 2019), and well-match observations in



terms of ice geometry, grounding-line position, and ice dynamics (Figures S8 and S9). In comparison to other ISMIP6 models, the root mean square error (RMSE) is within the range for both ice thickness (RMSE $\sim 50\,\mathrm{m}$) and ice surface velocity (RMSE 575 $\sim 100\,\mathrm{m\,yr^{-1}}$).

## Appendix B: PDD-based melt-and-runoff model

At the beginning of every year, monthly near-surface air temperatures and precipitation rates are used as inputs to a positive degree-day (PDD) algorithm that calculates the surface mass balance at the ice surface by capturing the fundamental physical processes of surface melting of ice and refreezing versus runoff in the snow column (Huybrechts and de Wolde, 1999; 580 Seguinot, 2013). More specifically, similar to Tsai et al. (2020), the algorithm involves seasonal cycles of zero-dimensional bulk quantities of snow and embedded meltwater, run through several years to equilibrium with a weekly time step, driven by seasonal variations of the air temperatures and precipitation rate interpolated in time to those time steps. The PDD scheme calculates the melt of snow or exposed ice at each weekly timestep (with a uniform normal distribution of standard deviation $\sigma_{\mathrm{PDD}} = 4\,°\mathrm{C}$ around the monthly mean $T_m$, representing diurnal cycles and synoptic variability) while tracking the evolving 585 thickness of the snow layer across the balance year. Surface melt is proportional to the number of positive degree days, using distinct coefficients $K_{\mathrm{snow}}$ and $K_{\mathrm{ice}}$ of melt per degree (C) day for snow and ice, respectively. Accumulation is assumed to equal precipitation when the daily near-surface temperature (also assumed to have a normal distribution around the monthly mean, using a smaller standard deviation of $3.5\,°\mathrm{C}$ to account for the smaller variations in temperature during cloudy days when precipitation occurs) is below $0\,°\mathrm{C}$, and decreasing linearly with temperature between 0 and $2\,°\mathrm{C}$ (above which precipitation 590 is then interpreted as rain; Seguinot, 2013). After seasonal equilibrium is reached, net annual quantities are used to calculate the refreezing of meltwater and runoff of excess meltwater once the snow is saturated. We use the approach proposed by Huybrechts and de Wolde (1999) based on a simple thermodynamic parameterisation of the refreezing process, i.e., the condition for refreezing depends on the cold content of the upper ice sheet layers. Therefore, the maximum amount of refreezing is given by $P_{\mathrm{ref}} = \frac{c_{\mathrm{i}}}{L_{\mathrm{f}}} d_{\mathrm{ice}}(T_{\mathrm{melt}} - T_{\mathrm{year}})$, where $c_{\mathrm{i}}$ and $L_{\mathrm{f}}$ are the specific heat capacity and latent heat of fusion of ice, respectively, 595 $T_{\mathrm{year}}$ is the annual mean temperature, $T_{\mathrm{melt}}$ is the melting point, and $d_{\mathrm{ice}}$ is the thickness of the thermally-active layer.

It is important to note that since the melt-and-runoff model is not used during the initialisation procedure (see Appendix A) SMB anomalies derived from the PDD-based melt-and-runoff model are used instead of absolute SMB values in order to maintain the steady-state with respect to initial (1950 CE) atmospheric conditions under unforced conditions. These anomalies are calculated with respect to the SMB reproduced by the melt-and-runoff scheme under the mean 1945-1955 air temperature 600 and precipitation conditions. Note also that the melt-and-runoff model described above has been improved compared with the PDD scheme implemented in previous model versions (f.ETISh; Pattyn, 2017; Bulthuis et al., 2019; Coulon et al., 2021, see Coulon et al., in prep.).



**Appendix C: More details on the applied sub-shelf melt parameterisations**

**C1    PICO model**

The PICO model is a box parameterisation developed by Reese et al. (2018a) based on the analytical steady-state solution of
the box model by Olbers and Hellmer (2010). Originally designed for a 2-D cavity, the model represents the buoyancy-driven
advection of ambient ocean water into the ice-shelf cavity up to the grounding line, and then upward along the ice draft through
consecutive boxes. The melt rates in each box are given by the following equation:

$$m_k = \gamma_T^\star \times \left(\frac{\rho_{\mathrm{w}} c_{\mathrm{p}}}{\rho_{\mathrm{i}} L_{\mathrm{f}}}\right) \times (T_k - T_{\mathrm{f},k}), \tag{C1}$$

where $\rho_{\mathrm{w}}$ and $c_{\mathrm{p}}$ are the density and specific heat of seawater, while $\rho_{\mathrm{i}}$ and $L_{\mathrm{f}}$ are the density and the latent heat of fusion
of ice. The subscript $k$ denotes properties evaluated in each box, with $T$ and $T_{\mathrm{f}}$ representing the temperatures of the ocean
and the freezing point, respectively. These properties account for the transformation of ocean temperature and salinity in
consecutive boxes through heat and salt turbulent exchange across the ocean boundary layer underneath ice shelves, driven by
ocean temperature and salinity near the seafloor. The effective turbulent temperature exchange velocity $\gamma_T^\star$ is assumed to be
constant and uniform. Reese et al. (2018a) calibrated the heat exchange and overturning coefficients to obtain realistic average
melt rates for the Pine Island and Ronne-Filchner ice shelves. Here, we adopt the overturning coefficient used by Reese et al.
(2018a) and vary the value of the effective heat exchange velocity $\gamma_T^\star$ within the range explored by Reese et al. (2018a), i.e.,
$0.1 \times 10^{-5} - 10 \times 10^{-5}$, hence including their best-fit value of $2 \times 10^{-5}$ m s$^{-1}$. The division into boxes for each ice shelf
follows the approach of Reese et al. (2018a).

Unlike the simple and plume parameterisations (see below), the PICO model does not use the vertical profile of ocean
properties as input. Reese et al. (2018a) consider the average properties of the ocean water in front of the ice-shelf cavities at
the depth of the continental shelf, obtained by averaging the observed properties of the Antarctic Shelf Bottom Water on the
continental shelves from Schmidtko et al. (2014) over larger basins. The far-field water then mixes with meltwater and rises
along the ice-shelf base due to buoyancy. In this study, when applying the 3-D present-day ocean climatology from Jourdain
et al. (2020), only the far-field properties at the average entrance depth of each ice-shelf cavity are advected to the grounding
line, following Burgard et al. (2022).

**C2    Plume model**

The plume model is a basal melt rate parameterisation based on the theory of buoyant meltwater plumes travelling upward
along the base of the ice shelf from the grounding line, driving the overturning circulation within the ice-shelf cavity. This
two-dimensional formulation by Lazeroms et al. (2018, 2019) emulates the behaviour of the one-dimensional plume model
developed by Jenkins (1991) for a plume travelling in an ocean with ambient temperature $T_a$ and salinity $S_a$ provided from
far-field ocean measurements (taken at the cavity scale, i.e., extrapolated to the ice draft depth for each point and then averaged
over the ice-shelf area; Lazeroms et al., 2019; Burgard et al., 2022). A first configuration of the plume model was proposed by



Lazeroms et al. (2018). Here, we applied the revised, more physical, version described in Lazeroms et al. (2019), where the
sub-shelf melt rates are given by

$$m = M_1 \times M_2 \times \frac{\rho_{\mathrm{w}}}{\rho_{\mathrm{i}}}, \tag{C2}$$

with $M_1$ and $M_2$ computed as follows:

$$M_1 = \left[ \frac{\beta_S S_a g}{\lambda_3 (L_{\mathrm{f}}/c_{\mathrm{p}})^3} \right]^{1/2} \left[ \frac{1 - c_{\rho 1} C_{\mathrm{d}}^{1/2} \Gamma_{TS}}{C_{\mathrm{d}} + E_0 \sin\theta} \right]^{1/2} \left[ \frac{C_{\mathrm{d}}^{1/2} \Gamma_{TS} E_0 \sin\theta}{C_{\mathrm{d}}^{1/2} \Gamma_{TS} + c_\tau + E_0 \sin\theta} \right]^{3/2} (T_a - T_{\mathrm{f,gl}})^2, \tag{C3}$$

and

$$M_2 = \frac{1}{2\sqrt{2}} [3(1-x)^{4/3} - 1] \sqrt{1 - (1-x)^{4/3}}, \tag{C4}$$

where $\beta_S$ is the salt contraction coefficient, $g$ the gravitational acceleration, $\lambda_3$ the liquidus pressure coefficient, $C_{\mathrm{d}}$ the drag
coefficient, $E_0$ the entrainment coefficient, $C_{\mathrm{d}}^{1/2} \Gamma_{TS}$ the effective thermal Stanton number, $\theta$ the local slope of the ice-shelf
base relative to the horizontal, and $T_{\mathrm{f,gl}}$ is the freezing temperature at the grounding line. The characteristic length scale $x$ is
defined as

$$x = \lambda_3 \frac{z_{\mathrm{draft}} - z_{\mathrm{gl}}}{T_{\mathrm{a}} - T_{\mathrm{f,gl}}} \left[ 1 + C_\epsilon \left( \frac{E_0 \sin\theta}{C_{\mathrm{d}}^{1/2} \Gamma_{TS} + c_\tau + E_0 \sin\theta} \right)^{3/4} \right]^{-1}, \tag{C5}$$

with $z_{\mathrm{draft}}$ and $z_{\mathrm{gl}}$ the local depths of the ice draft and the grounding line, respectively. For more details on the definitions
of the $c_{\rho 1}$ and $c_\tau$ coefficients and the values of the various parameters, please refer to Lazeroms et al. (2019). In this study, the
only parameter varied is the effective thermal Stanton number $C_{\mathrm{d}}^{1/2} \Gamma_{TS}$. We explore values between $1 \times 10^{-4}$ and $10 \times 10^{-4}$,
while the value used in Lazeroms et al. (2019) is $5.9 \times 10^{-5}$.

**C3  Local quadratic parameterisation**

In contrast to the more complex models described above, several simple parameterisations assume that the thermal forcing
across the ice–ocean boundary layer can be directly determined from far-field ocean conditions (Favier et al., 2019; Jourdain
et al., 2020; Burgard et al., 2022). The cooling of the water as it is transported from the far field into the ice-shelf cavity and
then mixed into the ice–ocean boundary layer is therefore simply accounted for through the choice of an effective heat transfer
coefficient (Favier et al., 2019). When assuming a quadratic, local dependency on the thermal forcing, the formulation of the
sub-shelf melting can be written as follows:

$$m(x,y) = \gamma_T \times \left(\frac{\rho_{\mathrm{w}} c_{\mathrm{p}}}{\rho_{\mathrm{i}} L_{\mathrm{f}}}\right)^2 \times \mathrm{TF}(x,y,z_{\mathrm{draft}})^2, \tag{C6}$$



where $\mathrm{TF} = T - T_{\mathrm{f}}$ is the thermal forcing at the ice–ocean interface and $\gamma_T$ is the heat exchange velocity. In this study, we vary $\gamma_T$ within the range of $1 \times 10^{-4} - 10 \times 10^{-4}\ \mathrm{m\,s^{-1}}$.

## C4   ISMIP6 non-local quadratic parameterisations

In contrast to the local dependency on the thermal forcing, non-local quadratic parameterisations consider that the local circulation at a draft point is influenced not only by local thermal forcing but also by its average over the ice-shelf basal surface (Favier et al., 2019). In the ISMIP6 standard approach developed by Jourdain et al. (2020), basal melt rates beneath ice shelves are computed using a slightly modified version of the non-local quadratic function of thermal forcing described in Favier et al.

(2019), with a regional thermal forcing correction:

$$m(x,y) = \gamma_0 \times (\frac{\rho_{\mathrm{w}} c_{\mathrm{p}}}{\rho_{\mathrm{i}} L_{\mathrm{f}}})^2 \times (\mathrm{TF}(x,y,z_{\mathrm{draft}}) + \delta T_{\mathrm{sector}}) \times |\langle \mathrm{TF}\rangle_{\mathrm{draft}\in\mathrm{sector}} + \delta T_{\mathrm{sector}}|, \tag{C7}$$

where $\langle \mathrm{TF}\rangle_{\mathrm{draft}\in\mathrm{sector}}$ represents the thermal forcing averaged over all the ice shelves of an entire sector. The uniform coefficient $\gamma_0$ (with units of velocity) is an empirical effective heat transfer coefficient. Sector-specific temperature corrections $\delta T_{\mathrm{sector}}$ are applied to match observation-based melt rates (at the scale of a sector).

Jourdain et al. (2020) also proposed a non-local quadratic parameterisation with a dependency on the local slope. It is obtained by multiplying the above equation by $\sin\theta$. As $\sin\theta$ is typically on the order of $10^{-2}$ near grounding lines, the value of $\gamma_0$ is increased by a factor of about 100. In our study, we vary $\gamma_0$ within the ranges of $1 \times 10^4 - 4 \times 10^4$ and $1 \times 10^6 - 4 \times 10^6$ $\mathrm{m\,yr^{-1}}$ for the non-local quadratic parameterisations without and with a dependency on the local slope, respectively. As a comparison, the median values determined by Jourdain et al. (2020) using their *MeanAnt* method (i.e., chosen so that the

total Antarctic melt rates given by the parameterisations match observational estimates) are $1.45 \times 10^4$ and $2.06 \times 10^6\ \mathrm{m\,yr^{-1}}$, respectively.

For both quadratic non-local parameterisations (with and without the dependency on the local slope), Jourdain et al. (2020) proposed sector temperature corrections for given values of $\gamma_0$. Here, we apply (for each parameterisation) the sector temperature corrections derived for the *MeanAnt* median $\gamma_0$ values, which lie approximately in the middle of our $\gamma_0$ uncertainty ranges.

For consistency, although these temperature corrections were derived using the ocean data provided by Jourdain et al. (2020), we also applied them when using the present-day ocean temperature and salinity fields based on the observed properties of the Antarctic Shelf Bottom Water on the continental shelves from Schmidtko et al. (2014). However, it is worth noting that this combination logically leads to lower calibration scores (see Figure S1).



**Appendix D: Estimates of Antarctic mass balance and its drivers over the past decades used for the assessment of the**
**calibrated ensemble**

**Table D1. Estimates of changes in Antarctic net mass balance, sub-shelf melt (basal mass balance), iceberg calving, and surface mass balance over the past decades from recent satellites-based and modelling studies, including uncertainties. These data were used to qualitatively assess the behaviour of the calibrated ensemble over the historical period (Figure 1).**

| Data Type | Study | Period | Value | Uncertainty | Unit |
|---|---|---|---|---|---|
| Net Mass Balance | Otosaka et al. (2023) | 1992-1996 | -70 | 40 | $\text{Gt yr}^{-1}$ |
| | | 1997-2001 | -19 | 39 | $\text{Gt yr}^{-1}$ |
| | | 2002-2006 | -62 | 41 | $\text{Gt yr}^{-1}$ |
| | | 2007-2011 | -130 | 45 | $\text{Gt yr}^{-1}$ |
| | Rignot et al. (2019) | 1979-1989 | -40 | 9 | $\text{Gt yr}^{-1}$ |
| | | 1989-1999 | -49.6 | 14 | $\text{Gt yr}^{-1}$ |
| | | 1999-2009 | -165.8 | 18 | $\text{Gt yr}^{-1}$ |
| | Bamber et al. (2018) | 1992-1996 | -27 | 106 | $\text{Gt yr}^{-1}$ |
| | | 1997-2001 | -103 | 106 | $\text{Gt yr}^{-1}$ |
| | | 2002-2006 | -25 | 54 | $\text{Gt yr}^{-1}$ |
| | | 2007-2011 | -117 | 28 | $\text{Gt yr}^{-1}$ |
| | Shepherd et al. (2018) | 1992-1997 | -49 | 67 | $\text{Gt yr}^{-1}$ |
| | | 1997-2002 | -38 | 64 | $\text{Gt yr}^{-1}$ |
| | | 2002-2007 | -73 | 53 | $\text{Gt yr}^{-1}$ |
| | | 2007-2012 | -160 | 50 | $\text{Gt yr}^{-1}$ |
| | | 1992-2011 | -76 | 59 | $\text{Gt yr}^{-1}$ |
| | Martín-Español et al. (2016) | 2010-2013 | -159 | 22 | $\text{Gt yr}^{-1}$ |
| Basal Mass Balance | Liu et al. (2015) | 2005-2011 | 1516 | 106 | $\text{Gt yr}^{-1}$ |
| | Depoorter et al. (2013) | 1995-2009 | 1454 | 174 | $\text{Gt yr}^{-1}$ |
| | Rignot et al. (2013) | 2003-2008 | 1500 | 235 | $\text{Gt yr}^{-1}$ |
| | Adusumilli et al. (2020) | 2003-2008 | 1500 | 140 | $\text{Gt yr}^{-1}$ |
| | | 1994-2018 | 1260 | 150 | $\text{Gt yr}^{-1}$ |
| Calving Fluxes | Liu et al. (2015) | 2005-2011 | 755 | 25 | $\text{Gt yr}^{-1}$ |
| | Depoorter et al. (2013) | 1995-2009 | 1321 | 144 | $\text{Gt yr}^{-1}$ |
| | Rignot et al. (2013) | 2003-2008 | 1089 | 139 | $\text{Gt yr}^{-1}$ |
| Surface Mass Balance | Mottram et al. (2021) | 1980-2010 | 2483 | 266 | $\text{Gt yr}^{-1}$ |
| | | 1987-2015 | 2329 | 94 | $\text{Gt yr}^{-1}$ |
| | Lenaerts et al. (2012) | 1979-2010 | 2418 | 181 | $\text{Gt yr}^{-1}$ |



*Code and data availability.* The code of Kori-ULB ice-sheet model is publicly available on GitHub via https://github.com/FrankPat/Kori-dev. All datasets used in this study are freely accessible through their original references. The CMIP6 forcing data used in this study are accessible through the CMIP6 search interface (https://esgf-node.llnl.gov/search/cmip6/). The MAR outputs used in this study are available on Zenodo (https://doi.org/10.5281/zenodo.4459259; Kittel et al., 2021). The simulations outputs, the data needed to produce the figures and
tables, and the scripts will be hosted on Zenodo upon acceptance of this manuscript.

*Author contributions.* VC and AKK conceived the study in collaboration with RW and FP. VC and AKK developed the experimental setup and design, with contributions from TE, FT and CK. VC and AKK processed the forcing data. VC set up the ice-sheet model and performed all model simulations. VC performed the data analysis, produced the figures, and wrote the original manuscript draft with regular inputs from AKK. All authors provided feedback on the analysis and input to the manuscript.

*Competing interests.* The authors declare that they have no conflict of interest.

*Acknowledgements.* This publication was supported by PROTECT. This project has received funding from the European Union's Horizon 2020 research and innovation programme under grant agreement No 869304, PROTECT contribution number xx. Computational resources have been provided by the Consortium des Équipements de Calcul Intensif (CÉCI), funded by the Fonds de la Recherche Scientifique de Belgique (F.R.S.-FNRS) under Grant No. 2.5020.11 and by the Walloon Region. Special thanks to Ariel Lozano for his relentless help and
unfailing availability. A.K.K. and R.W. acknowledge support by the European Union's Horizon 2020 research and innovation programme under Grant Agreement No. 820575 (TiPACCs) and No. 869304 (PROTECT).



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
