# Peer review of "Disentangling the drivers of future Antarctic ice loss with a historically-calibrated ice-sheet model"

_EGUsphere, 2023_

## Referee Comment (RC2)

**Review of "Disentangling the drivers of future Antarctic ice loss with a historically-calibrated ice sheet model", by Coulon, et al**

September 2023

**1 Overview**

In this work, the authors use a set of ice sheet models calibrated to match the historical record for the Antarctic Ice Sheet and run an ensemble of simulations under different climate scenarios to try to extract the drivers of future Antarctic ice loss and contribution to sea level rise. In particular, they examine the varied roles that oceanic and atmospheric forcing play in the different climate scenarios. I found that the experiment was well thought-out, the results are clearly presented, and the paper itself was clearly written. I believe that this work represents a significant advance and is suitable for publication after a few minor issues are addressed.

Overall, I am skeptical of the 16 km resolution used in this study. While the use of the Schoof criterion means that the method as a whole is somewhat impervious to resolution, there is some evidence that these sorts of approaches aren't the most accurate (i.e. [3] ). That said, I suspect the model is sufficient for the broad-strokes purposes of this study.

**2 Specific Points**

1. line 110: I think the first use of the term "committed sea level rise" was in Price [2]

2. Figure 2: The colormap used for probability is unfortunate in that the shading used to represent "no probability" is indistinguishable from the middle-scale shading (around 50% probability). Would a monochrome color scale make more sense here?

3. Figure 5: The "present-day" (control) experiment would be useful to include in this figure for comparison purposes.

4. Figure 6: It would be clearer if you point out the components which have opposite signs with regard to contribution to SLR (for example, surface

melt and runoff appear to be opposite signs). I think that using the conventions you're using makes it easier to tell when SMB components balance, but you should make that clear in the caption.

5. lines 350-360: You should probably also mention hydrofracture as an impact of changes in SMB on ice shelves (which is a mechanism via which atmospheric forcing can mimic ocean forcing and its impacts on buttressing).

6. line 555: Is the assumption of spatially constant viscoelastic properties appropriate? I'm not an expert, but I've seen a fair bit of recent work on how soft the bedrock under WAIS is relative to the rest of the AIS, and its impacts on ice sheet dynamics. Amusingly, I can cite work by Coulon, et al to make this point.[1] If you feel that the model used here is reasonable, I think you need to justify that given the existence of a body of work which seems to suggest otherwise.

7. line 686 (code availability) – I think you need to specify a particular version of the code. Specifying a dev branch won't be sufficient to fully reproduce the results here.

**3 Minor corrections and typos**

1. line 39: "compensate" – would "offset" be a better word here?

2. line 66: "allowing to quantify" → "allowing us to quantify"?

3. line 109: "allowing to investigate"...

4. line 111: "amount" → "number"

5. line 202: You use "Amundsen Sea Sector" here, while elsewhere you use "Amundsen Sea Embayment" – are they referring to the same region? If so, it would be better to be consistent.

6. Figure 1 caption: "area represent" → either "areas represent" or "area represents"

7. line 334: "Figure... illustrate"

8. line 379: Should the reference to Figure 2D here be to 2a?

9. Figure 7 caption: "aggregated fluxed" → "fluxes"

**References**

[1] Violaine Coulon, Kevin Bulthuis, Pippa L. Whitehouse, Sainan Sun, Konstanze Haubner, Lars Zipf, and Frank Pattyn. Contrasting response of West and East Antarctic Ice Sheets to glacial isostatic adjustment. *Journal of Geophysical Research: Earth Surface*, 126(7), 2021.

[2] S. F. Price, A. J. Payne, I. M. Howat, and B. E. Smith. Committed sea-level rise for the next century from Greenland ice sheet dynamics during the past decade. *Proceedings of the National Academy of Sciences*, 108(22):8978–8983, 2011.

[3] Ronja Reese, Ricarda Winkelmann, and Hilmar Gudmundsson. Grounding-line flux formula applied as a flux condition in numerical simulations fails for buttressed Antarctic ice streams. *The Cryosphere*, 12(10):3229–3242, October 2018.

---

## Author Comment (AC1)

**Author Response to Reviewer #1**

**Disentangling the drivers of future Antarctic ice loss with a historically-calibrated ice-sheet model**

Violaine Coulon, Ann Kristin Klose, Christoph Kittel, Tamsin Edwards, Fiona Turner, Ricarda Winkelmann, and Frank Pattyn
*The Cryosphere*, https://doi.org/10.5194/egusphere-2023-1532
* * *
**Reviewer Comment**,    Author Response,   *'changed manuscript text'*

**This paper presents a suite of new ice sheet simulations for Antarctica, that explore a range of critical parameters where substantial uncertainty exists. By using an observationally-constrained starting point and a rigorous statistical framework for assessing ensemble members, the authors are able to disentangle the dominant drivers of ice sheet evolution both through time (out to year 3000), and across different sectors of the continent. Their results are consistent with previous findings, in terms of contributions to sea level, but the paper makes a significant advance through its rigorous and multi-parameter approach. The figures are extremely good - both informative and clear - and the text is very well written.**

**Overall I could find nothing of much substance to comment on. There is a typo ('adressing') at line 114, but this is I think the only one I found. At line 424 I thought maybe the Robel and Banwell paper could be mentioned - in terms of how meltwater ponds might localise. At line 481 I thought maybe a comment about model resolution could be made, and followed up in more detail in the model description Appendix. I for one am not going to argue that a model resolution of 16 km is insufficient for this kind of study - I think it is entirely pragmatic for an ensemble approach like this - but I know that there are others in the community who might like to see more justification for a 'low res' simulation, or at least some evidence that the GL tracks reliably.**

**But these are minor points. Overall I found this a fascinating and enjoyable paper to read, and it will almost certainly be of great value for future assessments of SLR.**

Dear Nick Golledge,

Thank you very much for the positive feedback and for your constructive comments on how to further improve our manuscript. We plan to address your comments as follows:

- We corrected the typo at line 114, thank you for identifying it.
- We included Robel and Banwell (2019) as a reference in the discussion section. We have specified that the influence of cascades of interacting melt pond hydrofracture events, which has been shown by Robel and Banwell (2019) to limit the speed of ice-shelf collapse through hydrofracture processes, has been ignored here. Therefore, our projections may overestimate the risk of surface melt-induced destabilisation.
- With respect to the model resolution, we included the following statement in the discussion (line 481):

  *'As high spatial resolution remains a limiting factor for studying ice-sheet behaviour in an uncertainty quantification framework as presented here, we adopted a 16km*

*spatial resolution to allow for ensembles on multi-centennial timescales along with thorough parameter exploration. This relatively coarse resolution is used in combination with a flux condition allowing to account for grounding-line migration (as discussed in Appendix A). However, while our grounding-line migration may work effectively with coarser resolutions, it is important to note that smaller bedrock irregularities and pinning points (Morlighem et al., 2020) may well be overseen with this approach.'*

In addition, building upon your suggestion, we have introduced the following discussion regarding our use of a flux condition to account for grounding-line migration in Appendix A (line 538). The discussion is accompanied by a new figure (S12 in the supplementary material, shown below), which presents the results of the MISMIP+ Ice 1 experiments (Cornford et al., 2020) with Kori-ULB at different spatial resolutions (1, 2, and 4-km) with and without (at 2 and 4-km resolutions) the flux condition to determine grounding-line migrations.

*'Similarly, we find that applying a heuristic rule or parameterisation for the flux across the grounding line (Pattyn et al., 2017) passes the test of being able to maintain a steady state with the grounding line located on a retrograde slope due to buttressing (MISMIP+; Cornford et al., 2020, see Figure S12). In addition, it produces responses to the loss of the buttressing within the range of other ice-sheet models (using different ice-flow approximations), even at coarser resolutions (Figure S12; see discussion in Pollard and DeConto, 2020). Furthermore, multi-model ensemble estimates of future ice sheet response within ISMIP6 (Seroussi et al., 2020; Sun et al., 2020) clearly demonstrate that the overall behaviour of Kori-ULB (previously f.ETISh) is in line with results from the high-resolution models that participated in the ensemble.'*

We hope that these will constitute sufficient evidence that the grounding line tracks reliably and is within the range of other models using different ice-flow approximations.

[Figure]

Figure S12: **Grounding-line positions for the MISMIP+ Ice 1 experiments with Kori-ULB.** (a) Final grounding-line positions at the end of each step of the Ice 1 experiments within the MISMIP+ ice stream domain, which is a rectangular domain spanning 640 km in the $x$ direction and 80 km in the $y$ direction. Ice flows in a direction roughly parallel to the $x$ axis (with a mirror symmetry in the lateral center of the ice stream – midchannel position is at $y = 0$). (b) Midchannel grounding line positions plotted against time for the Ice 1 experiments. A retrograde bed slope is observed between $x = 400$ and $x = 500$ km. All experiments are run at different spatial resolutions (1, 2 and 4 km), and with (at 2 and 4-km resolution only) and without using a flux-condition to determine grounding-line migrations. Black curves and symbols correspond to the Ice 0 (control) experiment. Blue curves and symbols correspond to the Ice1r experiment (melt-induced retreat). Red curves and symbols correspond to the Ice1ra experiment (no melting readvance). Finally, yellow curves and symbols correspond to the Ice1rr experiment (further melt-induced retreat). More information on the experiments setup and the domain are provided in Cornford et al. (2020). Lines and shaded regions in (b) show the envelopes for the 'main subset' of MISMIP+ models, copied from Cornford et al. (2020, their Fig. 7a).

In addition to the above, we would like to draw your attention to a correction made to Figure 1. We initially displayed the ensemble mean and standard deviation while our intention was to present the median and 5-95% probability interval. The revised version of the manuscript includes the corrected figure (shown below).

[Figure]

Finally, we would like to inform you that the data acknowledgement section has been updated as follows to provide a Zenodo link (10.5281/zenodo.8398772) that grants access to the simulations outputs and associated scripts:

'*The code and reference manual of Kori-ULB ice-sheet model are publicly available on GitHub via https://github.com/FrankPat/Kori-dev. The specific Kori-ULB model version used in this study, the simulations outputs and the scripts needed to produce the figures and tables, and the scripts are hosted on Zenodo (https://doi.org/10.5281/zenodo.8398772). All datasets used in this study are freely accessible through their original references. The CMIP6 forcing data used in this study are accessible through the CMIP6 search interface (https://esgf-node.llnl.gov/search/cmip6/). The MAR outputs used in this study are available on Zenodo (https://doi.org/10.5281/zenodo.4459259; Kittel et al., 2021)*'

Best regards,

Violaine Coulon, on behalf of all co-authors.

**References:**

Cornford, S. L., Seroussi, H., Asay-Davis, X. S., Gudmundsson, G. H., Arthern, R., Borstad, C., Christmann, J., Dias dos Santos, T., Feldmann, J., Goldberg, D., Hoffman, M. J., Humbert, A., Kleiner, T., Leguy, G., Lipscomb, W. H., Merino, N., Durand, G., Morlighem, M., Pollard, D., Rückamp, M., Williams, C. R., and Yu, H. (2020). Results of the third Marine Ice Sheet Model Intercomparison Project (MISMIP+). *The Cryosphere*, 14(7):2283–2301. doi:10.5194/tc-14-2283-2020

Pattyn, F.: Sea-level response to melting of Antarctic ice shelves on multi-centennial timescales with the fast Elementary Thermomechanical Ice Sheet model (f.ETISh v1.0), *The Cryosphere*, 11, 1851–1878, https://doi.org/10.5194/tc-11-1851-2017, 2017.

Pollard, D. and DeConto, R. M.: Improvements in one-dimensional grounding-line parameterizations in an ice-sheet model with lateral variations (PSUICE3D v2.1), *Geosci. Model Dev.*, 13, 6481–6500, https://doi.org/10.5194/gmd-13-6481-2020, 2020.

Seroussi, H., Nowicki, S., Payne, A. J., Goelzer, H., Lipscomb, W. H., Abe-Ouchi, A., Agosta, C., Albrecht, T., Asay-Davis, X., Barthel, A., Calov, R., Cullather, R., Dumas, C., Galton-Fenzi, B. K., Gladstone, R., Golledge, N. R., Gregory, J. M., Greve, R., Hattermann, T., Hoffman, M. J., Humbert, A., Huybrechts, P., Jourdain, N. C., Kleiner, T., Larour, E., Leguy, G. R., Lowry, D. P., Little, C. M., Morlighem, M., Pattyn, F., Pelle, T., Price, S. F., Quiquet, A., Reese, R., Schlegel, N.-J., Shepherd, A., Simon, E., Smith, R. S., Straneo, F., Sun, S., Trusel, L. D., Van Breedam, J., van de Wal, R. S. W., Winkelmann, R., Zhao, C., Zhang, T., and Zwinger, T.: ISMIP6 Antarctica: a multi-model ensemble of the Antarctic ice sheet evolution over the 21st century, *The Cryosphere*, 14, 3033–3070, https://doi.org/10.5194/tc-14-3033-2020, 2020.

Sun, S., Pattyn, F., Simon, E., Albrecht, T., Cornford, S., Calov, R., . . . Zhang, T. (2020). Antarctic ice sheet response to sudden and sustained ice-shelf collapse (ABUMIP). *Journal of Glaciology, 66*(260), 891-904. doi:10.1017/jog.2020.67

---

## Author Comment (AC2)

**Author Response to Reviewer #2**

**Disentangling the drivers of future Antarctic ice loss with a historically-calibrated ice-sheet model**

Violaine Coulon, Ann Kristin Klose, Christoph Kittel, Tamsin Edwards, Fiona Turner, Ricarda Winkelmann, and Frank Pattyn
*The Cryosphere*, https://doi.org/10.5194/egusphere-2023-1532
* * *
**Reviewer Comment**,          Author Response,          '*changed manuscript text*'

**In this work, the authors use a set of ice sheet models calibrated to match the historical record for the Antarctic Ice Sheet and run an ensemble of simulations under different climate scenarios to try to extract the drivers of future Antarctic ice loss and contribution to sea level rise. In particular, they examine the varied roles that oceanic and atmospheric forcing play in the different climate scenarios. I found that the experiment was well thought-out, the results are clearly presented, and the paper itself was clearly written. I believe that this work represents a significant advance and is suitable for publication after a few minor issues are addressed.**

Dear Daniel Martin,

Thank you very much for your thoughtful review of our manuscript and your valuable feedback. Please find our detailed point-by-point responses (in blue) to your comments (in black) below.

**Overall, I am skeptical of the 16 km resolution used in this study. While the use of the Schoof criterion means that the method as a whole is somewhat impervious to resolution, there is some evidence that these sorts of approaches aren't the most accurate (i.e. [3]). That said, I suspect the model is sufficient for the broad-strokes purposes of this study.**

We appreciate your comment regarding the model resolution and the use of a flux condition in our study. In response to your concerns, we have added a discussion in Appendix A, supported by a new figure (S12 in the supplementary material, shown below), which presents the model's performance for the MISMIP+ Ice 1 experiments (Cornford et al., 2020) at different spatial resolutions, including 1, 2, and 4-km. This analysis includes simulations both with and without the flux condition to assess grounding-line migrations. We hope that this additional information provides confidence that our model reliably tracks grounding-line migration, and that it aligns with the range of responses seen in other ice-sheet models employing various ice-flow approximations. For further details on this topic and other modifications of the revised manuscript related to model resolution, please refer to our response to Reviewer #1.

[Figure]

Figure S12: **Grounding-line positions for the MISMIP+ Ice 1 experiments with Kori-ULB.** (a) Final grounding-line positions at the end of each step of the Ice 1 experiments within the MISMIP+ ice stream domain, which is a rectangular domain spanning 640 km in the $x$ direction and 80 km in the $y$ direction. Ice flows in a direction roughly parallel to the $x$ axis (with a mirror symmetry in the lateral center of the ice stream – midchannel position is at $y = 0$). (b) Midchannel grounding line positions plotted against time for the Ice 1 experiments. A retrograde bed slope is observed between $x = 400$ and $x = 500$ km. All experiments are run at different spatial resolutions (1, 2 and 4 km), and with (at 2 and 4-km resolution only) and without using a flux-condition to determine grounding-line migrations. Black curves and symbols correspond to the Ice 0 (control) experiment. Blue curves and symbols correspond to the Ice1r experiment (melt-induced retreat). Red curves and symbols correspond to the Ice1ra experiment (no melting readvance). Finally, yellow curves and symbols correspond to the Ice1rr experiment (further melt-induced retreat). More information on the experiments setup and the domain are provided in Cornford et al. (2020). Lines and shaded regions in (b) show the envelopes for the 'main subset' of MISMIP+ models, copied from Cornford et al. (2020, their Fig. 7a).

**2. Specific Points**

1. **line 110: I think the first use of the term "committed sea level rise" was in Price [2]**

   Thank you for your suggestion, we have included this reference.

2. **Figure 2: The colormap used for probability is unfortunate in that the shading used to represent "no probability" is indistinguishable from the middle-scale shading (around 50% probability). Would a monochrome color scale make more sense here?**

   Thank you for noting the colormap issue in Figure 2. We have revised the colormap to ensure clarity.

3. **Figure 5: The "present-day" (control) experiment would be useful to include in this figure for comparison purposes.**

   Thank you for your suggestion. We agree that including the "present-day" experiment for comparison is beneficial. However, to avoid clutter in Figure 5, we have displayed the mean thickness changes under constant present-day climate, SSP1-2.6 and SSP5-8.5 in Figure S4 in the supplementary material instead. We however added a statement in the caption of Figure 5 to guide readers to Figure S4 for comparison.

4. **Figure 6: It would be clearer if you point out the components which have opposite signs with regard to contribution to SLR (for example, surface melt and runoff appear to be opposite signs). I think that using the conventions you're using makes it easier to tell when SMB components balance, but you should make that clear in the caption.**

Many thanks for the relevant remark. We have clarified the caption of Figure 6 by adding the following sentence:

'*Note that positive SMB, snow accumulation and rainfall fluxes represent mass gains while positive surface melt and runoff fluxes represent mass losses.*'

5. **lines 350-360: You should probably also mention hydrofracture as an impact of changes in SMB on ice shelves (which is a mechanism via which atmospheric forcing can mimic ocean forcing and its impacts on buttressing).**

Thank you for pointing this out. We have added the following statement at line 358:

'*This reduction in ice-shelf buttressing with near-surface warming is further expected to be amplified by the influence of surface runoff on ice-shelf breakup through hydrofracturing (as demonstrated in section 3.5).*'

6. **line 555: Is the assumption of spatially constant viscoelastic properties appropriate? I'm not an expert, but I've seen a fair bit of recent work on how soft the bedrock under WAIS is relative to the rest of the AIS, and its impacts on ice sheet dynamics. Amusingly, I can cite work by Coulon, et al to make this point.[1] If you feel that the model used here is reasonable, I think you need to justify that given the existence of a body of work which seems to suggest otherwise.**

We agree that recent evidence indicates the presence of a low mantle viscosity and thin lithosphere beneath West Antarctica, contributing to a stabilizing potential of the West Antarctic ice sheet. However, it is important to note that these lateral variabilities in Antarctic viscoelastic properties are associated with considerable uncertainties, as discussed in Coulon et al. (2021). Therefore, and given the primary focus of our study on uncertainties related to ice-ocean-atmosphere interactions as drivers of mass loss, we have chosen not to explicitly incorporate this effect (and associated uncertainties). Instead, we opted for the common simplifying assumption of a uniform solid Earth representative of the continental average. In future work, which will involve applying a similar Bayesian calibration approach to a broader ensemble of simulations, we plan to address these uncertainties in a more comprehensive manner. This expanded scope will no longer be exclusively focused on ice-climate interactions, allowing for a more detailed exploration of additional uncertainties, including those related to solid Earth properties. In particular, we intend to explore uncertainties in intra-regional viscoelastic heterogeneities using a compilation of 3-D viscosity profiles of the Antarctic solid Earth (not yet publicly available).

To address this, in our revised manuscript, we have

- included the following statement in the discussion (line 478) that acknowledges the potential influence of this assumption:

  '*Accounting for the latter may therefore delay and/or reduce mass loss arising from West Antarctica (Whitehouse et al., 2019; Coulon et al., 2021). Future work will involve applying a similar Bayesian calibration approach to a broader ensemble of simulations sampling uncertainties no longer*

*exclusively focused on ice--climate interactions, allowing for a more detailed exploration of additional uncertainties.'*

- and added the following in Appendix A (line 556):

*',representative of the continental average (Le Meur and Huybrechts, 1996). We therefore did not account for the lateral variability nor the uncertainties in Antarctic viscoelastic properties (Whitehouse et al., 2019; Coulon et al., 2021).'*

7. **line 686 (code availability) – I think you need to specify a particular version of the code. Specifying a dev branch won't be sufficient to fully reproduce the results here.**

Thank you for your remark. The specific model version used in this study has been made available on Zenodo (10.5281/zenodo.8398772), in addition to access to the simulations outputs and associated scripts. The data acknowledgement section has therefore been updated as follows:

'*The code and reference manual of Kori-ULB ice-sheet model are publicly available on GitHub via https://github.com/FrankPat/Kori-dev. The specific Kori-ULB model version used in this study, the simulations outputs and the scripts needed to produce the figures and tables, and the scripts are hosted on Zenodo (https://doi.org/10.5281/zenodo.8398772). All datasets used in this study are freely accessible through their original references. The CMIP6 forcing data used in this study are accessible through the CMIP6 search interface (https://esgf-node.llnl.gov/search/cmip6/). The MAR outputs used in this study are available on Zenodo (https://doi.org/10.5281/zenodo.4459259; Kittel et al., 2021)*'

**3. Minor corrections and typos**
1. **line 39: "compensate" – would "offset" be a better word here?**

   Thanks, this has been changed.

2. **line 66: "allowing to quantify" → "allowing us to quantify"?**

   Thanks, this has been changed.

3. **line 109: "allowing to investigate"...**

   Thanks, this has been changed.

4. **line 111: "amount" → "number"**

   Thanks, this has been changed.

5. **line 202: You use "Amundsen Sea Sector" here, while elsewhere you use "Amundsen Sea Embayment" – are they referring to the same region? If so, it would be better to be consistent.**

   Yes, they are referring to the same region. Thanks for noticing, this has now been

changed.

6. **Figure 1 caption: "area represent" → either "areas represent" or "area represents"**

   Thanks, this has been changed.

7. **line 334: "Figure... illustrate"**

   Thanks, this has been changed.

8. **line 379: Should the reference to Figure 2D here be to 2a?**

   Indeed. Thanks, this has been changed.

9. **Figure 7 caption: "aggregated fluxed" → "fluxes"**

   Thanks, this has been changed.

In addition to the above, we would like to draw your attention to a correction made to Figure 1. We initially displayed the ensemble mean and standard deviation while our intention was to present the median and 5-95% probability interval. The revised version of the manuscript includes the corrected figure (shown below).

[Figure]

Best regards,

Violaine Coulon, on behalf of all co-authors.

**References:**

Coulon, V., Bulthuis, K., Whitehouse, P. L., Sun, S., Haubner, K., Zipf, L., and Pattyn, F.: Contrasting Response of West and East Antarctic Ice Sheets to Glacial Isostatic Adjustment, Journal of Geophysical Research: Earth Surface, 126, e2020JF006 003, https://doi.org/10.1029/2020JF006003, 2021.

Cornford, S. L., Seroussi, H., Asay-Davis, X. S., Gudmundsson, G. H., Arthern, R., Borstad, C., Christmann, J., Dias dos Santos, T., Feldmann, J., Goldberg, D., Hoffman, M. J., Humbert, A., Kleiner, T., Leguy, G., Lipscomb, W. H., Merino, N., Durand, G., Morlighem, M., Pollard, D., Rückamp, M., Williams, C. R., and Yu, H. (2020). Results of the third Marine Ice Sheet Model Intercomparison Project (MISMIP+). *The Cryosphere*, 14(7):2283–2301. doi:10.5194/tc-14-2283-2020

Le Meur, E. and Huybrechts, P.: A comparison of different ways of dealing with isostasy: examples from modelling the Antarctic ice sheet during the last glacial cycle, Annals of Glaciology, 23, 309–317, https://doi.org/10013/epic.12717.d001, 1996.

Whitehouse, P. L., Wiens, D. A., Gomez, N., and King, M. A.: Solid Earth change and the evolution of the Antarctic Ice Sheet, Nature Communications, pp. 1–14, https://doi.org/10.1038/s41467-018-08068-y, 2019.

---

## Author Response (AR2)

Dear Jan de Rydt,

Thank you again for handling the editing process of our manuscript and for your feedback.

We have now incorporated a more comprehensive discussion with respect to the choice of spatial resolution in the revised manuscript. We have undertaken a comparison between 16km and 8km pan-Antarctic constant present-day climate simulations, as suggested. The 4km pan-Antarctic run was not feasible for computational reasons. The results are presented in Supplementary Figure S13. Additionally, we have included results from 8km simulations in the MISMIP+ plot (Supplementary Figure S12) to offer further insights into the resolution-related aspects of our study.

The pan-Antarctic 8 – 16 km comparison shows that comparable mass changes occur in both runs in similar regions. The differences in mass change between both simulations lie well within the uncertainty bounds of our ensemble and may be attributed to the fact that both experiments represent a different bedrock topography (i.e., more detailed in the 8-km configuration).

We would like to underline that the goal of the MISMIP+ experiment serves a different purpose, i.e., it was carried out to demonstrate the validity/usefulness of the parameterisation for the flux across the grounding line, when grounding line migration fails at spatial resolutions that are too coarse. It is therefore not intended as a real convergence test. To address this concern, in addition to Figure S13, we are referencing Cornford et al. (2016) and Reese et al. (2018) more explicitly in our discussion. Especially, we emphasise the potential limitations of coarse resolution and the grounding line flux condition, as advised.

To enhance transparency, we have included an additional sentence in the discussion section, stating "We may therefore expect differences between our results and results at higher spatial resolutions, especially for small ice streams and outlets." This addition aims to underline our awareness of the limitations associated with the chosen resolution.

We hope that these modifications satisfactorily address your concerns about the justification for the 16km grid resolution for such a framework of producing ensembles of multi-centennial pan-Antarctic simulations.

Best regards,

   Violaine Coulon, on behalf of all co-authors.